



# Factors Controlling Black Carbon Distribution in the Arctic

Ling Qi[1,2], Qinbin Li[1,2], Yingrui Li[3], Cenlin He[1,2]

[1]Department of Atmospheric and Oceanic Sciences, University of California, Los Angeles, CA, USA
[2]Joint Institute for Regional Earth System Science and Engineering, University of California, Los Angeles, CA, USA
[3]School of Physics, Peking University, Beijing, China

*Correspondence to*: Ling Qi (qiling@atmos.ucla.edu)

**Abstract**. We investigate the sensitivity of black carbon (BC) in the Arctic, including BC in snow ($BC_{snow}$, ng g$^{-1}$) and surface air ($BC_{air}$, μg m$^{-3}$), to emissions, dry deposition and wet scavenging using a global 3-D chemical transport model (CTM) GEOS-Chem. We find that the model underestimates $BC_{snow}$ in the Arctic by 40% on average (median = 11.8 ng g$^{-1}$). Natural gas flaring
substantially increases total BC emissions in the Arctic (by ~70%). The flaring emissions lead to up to 49% increases (0.1−8.5 ng g$^{-1}$) in Arctic $BC_{snow}$, dramatically improving model comparison with observations (50% reduction in discrepancy) near flaring source regions (Western Extreme North of Russia). Ample observations suggest that BC dry deposition velocities over snow and ice in current CTMs (0.03 cm s$^{-1}$ in GEOS-Chem) are exceedingly small. We apply the resistance-in-series method to compute the dry deposition velocity that varies with local meteorological and surface conditions. The resulting velocity is
significantly larger and varies by a factor of eight in the Arctic (0.03–0.24 cm s$^{-1}$), increases the fraction of dry to total BC deposition (16% to 25%), yet leaves the total BC deposition and $BC_{snow}$ in the Arctic unchanged. This is largely explained by the offsetting higher dry and lower wet deposition fluxes. Additionally, we account for the effect of the Wegener-Bergeron-Findeisen (WBF) process in mixed-phase clouds, which releases BC particles from condensed phases (water drops and ice crystals) back to the interstitial air and thereby substantially reduces the scavenging efficiency of BC (by 43−76% in the Arctic).
The resulting $BC_{snow}$ is up to 80% higher, BC loading is considerably larger (from 0.25 to 0.43 mg m$^{-2}$), and BC lifetime is markedly prolonged (from 9 to 16 days) in the Arctic. Over all, flaring emissions increase $BC_{air}$ in the Arctic (by ~20 ng m$^{-3}$), the updated dry deposition velocity more than halves $BC_{air}$ (by ~20 ng m$^{-3}$), and the WBF effect increases $BC_{air}$ by 25−70% during winter and early spring. The resulting model simulation of $BC_{snow}$ is substantially improved (within 10% of the observations) and the discrepancies of $BC_{air}$ are much smaller during snow season at Barrow, Alert and Summit (from -67%−-47% to -46%−3%).
Our results point toward an urgent need for better characterization of flaring emissions of BC (e.g. the emission factors, temporal and spatial distribution), extensive measurements of both the dry deposition of BC over snow and ice, and the scavenging efficiency of BC in mixed-phase clouds.

## 1. Introduction

Black carbon (BC, loosely also known as soot), light absorbing refractory carbonaceous aerosols, influence climate through
direct absorption of solar radiation, semi-direct cloud effects, indirect cloud effects, and snow-albedo effect (Bond et al., 2013; IPCC, 2014). BC deposited on surfaces with high albedo, such as snow and ice, reduces surface albedo (the so-called snow albedo effect), increases surface solar heating, and accelerates snow and ice melting (Flanner et al., 2007, 2012; He et al., 2014b; Liou et al., 2014). This snow albedo feedback leads to enhanced BC radiative forcing (Bond et al., 2013 and references therein). Warren and Wiscombe (1985) highlighted the climate effect of fallen soot from fires started by nuclear explosions, which
reduced the surface reflectivity of snow and sea ice in the Arctic. Measurements by Clarke and Noone (1985) showed that there was ample amount of BC in the Arctic snow to exert climate impact in the region. Using observations of $BC_{snow}$, Hansen and





Nazarenko (2004) quantified, for the first time, the albedo reduction due to BC deposition on snow and ice (2.5% on average) across the Arctic. Snow albedo effect of BC in the Arctic has since received wide attention. Numerous studies have examined the snow albedo change in this region due to BC deposition (Jacobson, 2004; Marks and King, 2013; Namazi et al., 2015; Tedesco et al., 2016) and estimated the associated surface BC snow albedo radiative forcing to be substantial (0.024–0.39 W m$^{-2}$) in the

Arctic (Bond et al., 2013 and references therein; Flanner, 2013; Jiao et al., 2014; Namazi et al., 2015), comparable to the forcing of tropospheric ozone (0.37 W m$^{-2}$, Shindell et al., 2009). BC deposited on snow and ice is likely to be an important reason for unexpected rapid sea ice shrinkage in the Arctic (Koch et al., 2009; Goldenson et al., 2012; Stroeve et al., 2012). Widespread surface melting of the Greenland ice sheet was attributed to rising temperatures and reductions in surface albedo resulting from deposition of BC from northern hemispheric forest fires (Keegan et al., 2014; Tedesco et al., 2016).

To better constrain the radiative forcing and the associated uncertainty of BC snow albedo effect in the Arctic, it is imperative to improve the prediction of $BC_{snow}$ in the region. Previous studies found large discrepancies between modeled and observed $BC_{snow}$ (up to a factor of six) in the Arctic (e.g. Flanner et al., 2007; Koch et al., 2009). A comprehensive survey of $BC_{snow}$ observations across the Arctic (~1000 snow samples) by Doherty et al. (2010) provides a unique opportunity to constrain $BC_{snow}$ in the region.

Bond et al. (2013) compared results of $BC_{snow}$ from the Community Atmospheric Model (CAM version 3.1) (Flanner et al., 2009) and the Goddard Institute of Space Studies (GISS) model (Koch et al., 2009) with the observations from Doherty et al. (2010), averaged over the eight Arctic sub-regions (Fig. 1) as defined by Doherty et al. (2010). The resulting ratio of modeled to observed $BC_{snow}$ (sub-regional means) were 0.6−3.4 for CAM3.1 and 0.3−1.6 for GISS. Jiao et al. (2014) found large discrepancies in $BC_{snow}$ (up to a factor of six) between results from the Aerosol Comparisons between Observations and Models

(AeroCom, http://aerocom.met.no/) and the Doherty et al. (2010) observations. They also found large variations in BC deposition fluxes amongst the AeroCom models. Jiao et al. (2014) further pointed out that BC transport and deposition processes are more important for differences in simulated $BC_{snow}$ than differences in snow meltwater scavenging rates or emissions in models.

Studies have shown that Arctic atmospheric BC on average cools the surface due to surface dimming, while BC in the lower

troposphere warms the surface with a climate sensitivity (surface temperature change per unit forcing) of 2.8 ± 0.5 K W$^{-1}$ m$^2$ due to low clouds and sea-ice feedbacks that amplify the warming (e.g. Flanner, 2013). This sensitivity is a factor of two larger than that of BC snow albedo feedback (1.4 ± 0.7 K W$^{-1}$ m$^2$, Flanner, 2013), a factor of four larger than that of $CO_2$ (0.69 K W$^{-1}$ m$^2$, Bond et al., 2013) and much larger than that of tropospheric ozone (0.2 K W$^{-1}$ m$^2$, Shindell et al., 2009). However, estimates of BC in surface air (hereafter referred to as $BC_{air}$) in the Arctic are associated with large uncertainties (Textor et al., 2006, 2007;

Koch et al., 2009; Liu et al., 2011; Browse et al., 2012; Sharma et al, 2013). Observations have shown decreasing trends of $BC_{air}$ since 1990 at Barrow, Alert and Zeppelin (Sharma et al., 2004, 2006; Eleftheriadis et al., 2009). The modeling studies by Sharma et al. (2006, 2013) reproduced the declining trends and have associated these trends with declining emissions in Russia. In general, current models failed to reproduce the seasonal cycles of $BC_{air}$ observed at the aforementioned sites, with large underestimates during Arctic haze and overestimates in summer (Textor et al., 2006, 2007; Koch et al., 2009; Liu et al., 2011;

Browse et al., 2012; Sharma et al, 2013). The low biases are likely due to uncertainties associated with treatments of BC aging in the models (Liu et al., 2011; He et al., 2016), excessive dry deposition of BC (Huang et al., 2010; Liu et al., 2011) and wet scavenging of BC (Koch et al., 2009; Huang et al., 2010; Bourgeois and Bey, 2011; Liu et al., 2011), and overly efficient vertical mixing (Koch et al., 2009). Liu et al. (2011) used an OH dependent aging scheme, constrained a low dry deposition velocity and suppressed scavenging efficiency in ice clouds. They found that with all these improvements, the discrepancy of modeled and

simulated $BC_{air}$ during Arctic haze reduced from orders of magnitude to within a factor of two. Browse et al. (2012) suppressed





the scavenging of BC in ice clouds and included low level cloud scavenging and found an improved simulation of BC seasonal cycles in the Arctic. Recently, Huang et al. (2015) pointed out that the low biases are partly due to uncertainties in the estimates of BC emissions in Russia, which likely resulted in biases in both BC emission rates and spatial distributions.

In this study, we systematically investigate the controlling factors of BC distribution in snow and surface air in the Arctic using a 3-D chemical transport model GEOS-Chem. We first examine and incorporate gas flaring emissions of BC around the Arctic circle (Sect. 4.1). We then discuss and improve dry deposition velocity of BC in the Arctic (Sect. 4.2). Finally, we analyze BC wet scavenging efficiency (Sect. 4.3) and the sensitivity of $BC_{snow}$ and $BC_{air}$ to precipitation (Sect. 4.4).

## 2. BC observations in the Arctic

### 2.1 Measurements of BC in snow

The most comprehensive measurements $BC_{snow}$ were in eight sectors in the Arctic: Alaska, Arctic Ocean, Canadian Arctic, Canadian sub-Arctic, Greenland, Russia, Ny_Ålesund and Tromsø, mostly from March to May during 2005–2009 (Doherty et al., 2010; data available at http://www.atmos.washington.edu/sootinsnow/). Samples were for full snowpack depth and the sampling sites are shown in Fig. 1 (color coded by the sub-regions). These observations provide a reasonable constraint on

Arctic-wide annual-mean radiative effect from BC deposited in snow (Jiao et al., 2014).

Doherty et al. (2010) measured the light absorption of impurity in snow samples using the Integrating Sphere/Integrating Sandwich optical method and derived equivalent, maximum, and estimated $BC_{snow}$ using the wavelength-dependent absorption of BC and non-BC fractions (Doherty et al., 2010). We use here the estimated $BC_{snow}$. The largest sources of uncertainty stem from

uncertainties of BC mass absorption cross-section (MAC), BC absorption Ångstrom exponent ($Å_{BC}$), and non-BC absorption Ångstrom exponent ($Å_{non-BC}$) constituents. Doherty et al. (2010) used MAC = 6.0 $mg^2\ g^{-1}$ (at 500 nm), the MAC of their calibration filters. Using MAC = 7.5 $mg^2\ g^{-1}$ (at 500 nm) as recommended by Bond and Bergstrom (2006) would increase the estimated $BC_{snow}$ by ~25%. Doherty et al. (2010) used $Å_{BC}$ = 1.0 (range: 0.8–1.9) and $Å_{non-BC}$ = 5.0 (range: 3.5–7.0) in their derivation and estimated a 50% error in the estimated $BC_{snow}$. Additional uncertainties include instrumental uncertainty ($\leq$ 11%),

under-catch correction ($\pm$15%), and loss of aerosol to plastic flakes in the collection bags ($\pm$20%) for samples from West Russia and the Canadian sub-Arctic. The overall uncertainty of the estimated $BC_{snow}$ is < 60%.

### 2.2 Measurements of BC in surface air

In-situ measurements of $BC_{air}$ from 2007 to 2009 are available at five stations within the Arctic Circle (Fig. 1): Denali, AL (63.7°N, 149.0°W, 0.66 km a.s.l.), Barrow, AL (71.3°N, 156.6°W, 0.01 km a.s.l.), Alert, Canada (82.3°N, 62.3°W, 0.21 km

a.s.l.), Summit, Greenland (72.6°N, 38.5°W, 3.22 km a.s.l.), and Zeppelin, Norway (79°N, 12°E, 0.47 km a.s.l.). Data descriptions are shown in Table 1. Denali is part of the Interagency Monitoring of PROtected Visual Environment (IMPROVE) network (Malm et al., 1994; data available at http://vista.cira.colostate.edu/improve/). IMPROVE measurements are made every three days and 24-hour averages are reported. Thermal Optical Reflectance (TOR) combustion method is used based on the preferential oxidation of organic carbon (OC) and BC at different temperatures (Chow et al., 2004). BC-like products of OC

pyrolysis can lead to an overestimate of the BC mass. The uncertainties of the TOR method are difficult to quantify (Park et al., 2003; Chow et al., 1993).



Barrow is part of the NOAA Global Monitoring Division (GMD) network, where BC light absorption coefficients are measured from a particle soot absorption photometer (PSAP) since 1997 (Bond et al, 1999; Delene and Ogren, 2002; data available at http://www.esrl.noaa.gov/gmd/aero/net/). PSAP measures the change in light transmission at three wavelengths (467, 530 and 660 nm) through filter on which particles are collected. We used the measurements at 530 nm in this study. Site Barrow is about
8 km northeast of the village of Barrow and has a prevailing east-northeast wind off the Beaufort Sea. It is receives minimal influence from anthropogenic effects. The Arctic Ocean is less than 3 km northwest of the site. Because of its proximity to these bodies of water and the fact that the prevailing winds are off the Beaufort Sea, station Barrow is perhaps best characterized as having an Arctic maritime climate affected by variations of weather and sea ice conditions in the Central Arctic.

$BC_{air}$ at Alert were measured using an aethalometer model AE-6 with 1-wavelength operated by Environment Canada (Sharma et al., 2004; 2006; 2013; data available at http://www.ec.gc.ca/). The instruments measure the attenuation of light transmitted through particles that accumulate on a quartz fiber filter at 880nm. Alert station is located the furthest north of all the Arctic stations on the north-eastern tip of Ellesmere Island. The surroundings, both land and ocean, are mainly ice or snow covered during 10 months of the year (Hirdman et al., 2010). Alert is the station most isolated from continental source regions due to its
location deep within the Arctic.

The Zeppelin observatory is part of the European Supersites for Atmospheric Aerosol Research, where BC mass concentrations are also measured by an aethalometer and reported for seven wavelengths (370, 470, 520, 590, 660, 880 and 950 nm) (Eleftheriadis et al., 2009; data available at http://ebas.nilu.no/). We use the 520 nm data. Zeppelin is situated on a mountain
ridge on the western coast of Spitsbergen. Comparison of the sea level observations showed that the aerosol measurements at Zeppelin may be considered to be generally representative of free troposphere conditions (Eleftheriadis et al., 2009).

BC mass concentrations were also measured by an aethalometer at Summit (von Schneidemesser et al., 2009; data available at http://www.esrl.noaa.gov/gmd/aero/net/ ). Summit is located on the top of the Greenland glacial ice sheet, and surrounded by
very flat and homogeneous terrain for more than 100km in all directions (Hirdman et al., 2010). Moreover, measurements at Summit are representative for the Arctic free troposphere, due to the high elevation (3.2 km).

The uncertainty of filter-based absorption measurements of BC (PSAP and aethalometer) lies in empirical corrections of the overestimated absorption if light transmission is also affected by particulate light scattering (Bond et al., 1999). Accuracy of this
correction is 20–30% (Delene and Ogren, 2002; Weingartner et al., 2003; Virkkula et al., 2005). Additional uncertainty results from the empirical conversion from optical response to BC mass using an assumed mass absorption cross-section (MAC), which depends on the composition and morphology of the particles used in the calibration of the instrument and on the specific technique used to quantify the BC mass (Clarke et al., 1987; Slowik et al., 2007). The MAC of BC varies by up to a factor of four, from 5 $m^2$ $g^{-1}$ in remote areas to 20 $m^2$ $g^{-1}$ near source regions (Weingartner et al., 2003). We use 9.5 $m^2$ $g^{-1}$ for station
Barrow at wavelength 530 nm as recommended for the ARCTAS period (McNaughton et al., 2011; Wang et al., 2011). The MAC used at station Alert (Sharma et al., 2013), Zeppelin (Eleftheriadis et al., 2009) and Summit are 19 $m^2$ $g^{-1}$, 15.9 $m^2$ $g^{-1}$. The uncertainty of absorption enhancement by non-BC absorbers (organic carbon and mineral dust) is generally difficult to quantify unless the non-BC absorbers contribute more than 40% of absorption (Petzold et al., 2013).





## 3. Model description and simulations

### 3.1 GEOS-Chem simulation of BC

GEOS-Chem is a global 3-D chemical transport model driven with assimilated meteorology from the Goddard Earth Observing System (GEOS) of the NASA Global Modeling and Assimilation Office (GMAO). GEOS-5 meteorological data set are used to

drive model simulation at 2° lat × 2.5° lon resolution and 47 vertical layers from the surface to 0.01 hPa. Tracer advection is computed every 15 min with a flux-form semi-Lagrangian method (Lin and Rood, 1996). Tracer moist convection is computed using GEOS convective, entrainment, and detrainment mass fluxes as described by Allen et al. (1986a, b). Deep convection is parameterized using the relaxed Arakawa-Schubert scheme (Moorthi and Suarez, 1992; Arakawa and Schubert, 1974), and the shallow convection treatment follows Hack (1994). BC aerosols are emitted by incomplete fossil fuel and biofuel combustion

and biomass burning. We use global BC emissions from Bond et al. (2007) with updated emissions in Asia from Zhang et al. (2009). Biomass burning emissions are from the Global Fire Emissions Database version 3 (GFEDv3) (van der Warf et al., 2010) with updates for small fires in Randerson et al. (2012). It is assumed that 80% of the freshly emitted BC aerosols are hydrophobic (Park et al., 2003) and are converted to hydrophilic with an e-folding time of 1.15 days, which yields a good simulation of BC export efficiency in continental outflow (Park et al., 2005). Dry deposition in the model is computed using a

resistance-in-series method (Wesely, 1989; Zhang et al., 2001), whereas it assumes a constant aerosol dry deposition velocity of $0.03$ cm s$^{-1}$ over snow and ice (see Sect. 3.3). Wet deposition follows Liu et al. (2001), with updates as described in Wang et al. (2011).

### 3.2 Gas flaring emissions of BC

Gas flaring is a widely used practice for the disposal of natural gas in petroleum producing areas, where there is insufficient

infrastructure to make use of the gas (Elvidge et al., 2009; 2011). As such, gas flaring is widely recognized as a waste of energy and an added load of carbon emissions to the atmosphere. Stohl et al. (2013) derived BC emissions from gas flares in the oil and gas industry based on the gas flaring volumes developed within the Global Gas Flaring Reduction initiative and emission factors derived on the basis of particulate matter and soot estimates from CAPP (2007); John- son et al. (2011) and US EPA (1995). The flaring volumes were estimated using low light imaging data acquired by the Defense Meteorological Satellite Program (DMSP)

(Elvidge et al., 2011). The DMSP estimates of flared gas volume are based on a calibration developed with a pooled set of reported national gas flaring volumes and data from individual flares. The resulting gas flaring emissions (228 Gg yr$^{-1}$) accounts for ~5% of global anthropogenic emissions (4.8 Tg yr$^{-1}$, Bond et al., 2007) and ~3% of global total emissions (8.5 Tg yr$^{-1}$, including anthropogenic emissions from Bond et al., 2007 and Zhang et al., 2009 and biomass burning emissions from Randerson et al., 2012). However, the largest contributor Russia, contributing ~30% to the global flaring volume, locates in the

clean Arctic Circle. About 40% of BC emissions in the Arctic (115 Gg yr$^{-1}$) are from gas flaring (48 Gg yr$^{-1}$), shown in Fig. 1. It is estimated that flaring emissions contribute 42% to the annual mean BC$_{air}$ at surface in the Arctic (Stohl et al., 2013). However, to our knowledge, no study so far has investigated the contribution of flaring emissions to BC$_{snow}$ in the Arctic. Thus, we included flaring emissions from Stohl et al. (2013, data on flaring emissions is available at http://eclipse.nilu.no upon request) and investigated the contribution of flaring emissions to BC$_{snow}$ and BC$_{air}$ in the Arctic in Experiment B (Table 2).

### 3.3 Dry deposition over snow and ice

Nilsson and Rannik (2001) conducted eddy-covariance flux measurements of aerosol number dry deposition in the Arctic Ocean and found a mean dry deposition velocity ($v_d$) of 0.19 cm s$^{-1}$ over open sea, 0.03 cm s$^{-1}$ over ice floes and 0.03−0.09 cm s$^{-1}$ over



leads (Table 3). Following Nilsson and Rannik (2001), Fisher et al. (2011) imposed $v_d = 0.03$ cm s$^{-1}$ for aerosols over snow and ice. They found improved agreements of simulated sulfate with in-situ observations in spring and winter in the Arctic. Wang et al. (2011), also imposing $v_d = 0.03$ cm s$^{-1}$ for aerosols over snow and ice, and found better agreements for BC at the same stations as used by Fisher et al. (2011). They thus recommended a uniform $v_d = 0.03$ cm s$^{-1}$ for sulfate and BC over snow and ice. To

capture the winter and spring haze, other studies also used relatively low $v_d = 0.01$–$0.07$ cm s$^{-1}$ (Liu et al., 2011; Sharma et al., 2013). These low values, however, are likely too small for snow-covered land surface, where larger roughness lengths reduce the aerodynamic resistance thereby increase $v_d$ (Gallagher et al., 2002). The roughness length is 0.005 m for sea ice and 0.03−0.25 m for snow-covered land surface with grass and scattered obstacles (Wieringa, 1980). As a result, $v_d$ is larger over snow-covered land surface than over sea ice. Observed values over snow and ice are 0.01–2.4 cm s$^{-1}$ for aerosol particles in general and 0.01–

1.52 cm s$^{-1}$ for BC in particular (Table 3). Again, this suggests that a uniform value of $v_d = 0.03$ cm s$^{-1}$ is problematic. We apply the resistance-in-series method to calculate dry deposition velocity of BC over snow and ice, as a function of aerodynamic resistance, particle density and size and surface types (Experiment C, Table 2).

We would like to note that most of these observations (Held et al., 2011; Nilsson and Rannik, 2001; Bergin et al., 1995) were

from summertime Arctic (June−August) and clean regions (e.g., the Arctic and Greenland) far from anthropogenic pollutions. In addition, most of the dry deposition velocity measurements are for general aerosol particles. The only available dry deposition velocities specific to BC particles are derived from the strong surface enhancement of BC$_{snow}$ between two snow events at Mt. Changbai (42.5°N, 128.5°E, 0.74 km) in Northern China (Table 3). Wang et al. (2014) derived $v_d = 0.16$−$1.52$ cm s$^{-1}$. They then calculated dry deposition fluxes of $2.65 \pm 1.93$ μg cm$^{-2}$ month$^{-1}$, a factor of five higher than the corresponding wet deposition

fluxes. Despite these uncertainties, these measurements suggest that the low dry deposition velocity used in previous studies (Fisher et al., 2010; Liu et al., 2011; Wang et al., 2011; Sharma et al., 2013) might underestimate the role of dry deposition during snow season, particularly near source regions. Wang et al. (2014) concluded that dry deposition in the boundary layer may dominate over wet deposition during dry season in some regions, particularly near source regions with high BC$_{air}$. It is thus imperative to obtain measurements of BC dry deposition velocities in polluted regions in Russia and Northern Europe in spring,

when radiative forcing associated with BC snow-albedo effect is maximum (Flanner et al., 2009, 2012).

### 3.4 Wegener-Bergeron-Findeisen (WBF) process in mixed-phase clouds

Most AeroCom models (Textor, 2006) parameterize rainout rate following Giorgi and Chameides (1986). The rainout ratio is proportional to precipitation formation rate and mass mixing ratio of BC in condensed phase in clouds, which is determined by the scavenging efficiency of BC ($r_{scav.}$),

$$r_{scav.} = \frac{[BC]_{condensed}}{[BC]_{interstitial} + [BC]_{condensed}} \, , \qquad (1)$$

where $r_{scav.}$ is the scavenging efficiency and quantifies the partition of BC aerosols between condensed phase and the interstitial air; $[BC]_{condensed}$ is the mass mixing ratio of BC in condensed phase, including water drops and ice crystals in clouds, $[BC]_{interstitial}$ is the mass mixing ratio of BC in the interstitial air.

Hygroscopicity and size of BC-containing particles are determining factors for BC scavenging efficiency (Sellegri et al., 2003; Hallberg et al., 1992, 1995). Internal mixing with soluble inorganic species enhances the scavenging efficiency for aged BC particles (Sellegri et al., 2003). The scavenging efficiency is $0.39 \pm 0.16$ for BC-containing particles with diameter smaller than





0.3 μm and a small fraction (38%) of soluble inorganic material. The scavenging efficiency increases to 0.97±0.02 for particles with diameter larger than 0.3 μm and a larger fraction (57%) of soluble inorganic material (Sellegri et al., 2003). In addition to particle properties, cloud microphysics and dynamics play a significant role in determining the scavenging efficiency of BC in mixed-phase clouds (Hitzenberger et al., 2000, 2001; Cozic et al., 2007; Hegg et al., 2011). Measured BC scavenging efficiency

decreased from 0.60 in liquid only clouds to 0.05–0.10 in mixed-phase clouds, a reduction of more than a factor of five (Cozic et al., 2007; Henning et al., 2004; Verheggen et al., 2007). Such reduction was attributed to the effect of the WBF process (Cozic et al., 2007). In mixed-phase clouds, ice crystals grow at the expense of water drops when the environmental vapor pressure is higher than the saturation vapor pressure of ice crystals but lower than the saturation vapor pressure of water droplets (Wegener, 1911; Bergeron, 1935; Findeisen, 1938). As such, BC-containing particles in the water drops are released back to the interstitial

air and consequently the scavenging efficiencies are reduced. Another process, riming (Hegg et al., 2011), in mixed-phase clouds has an opposite effect to BC scavenging. When ice particles fall and collect the water drops along the pathway, the snow particles show rimed structure and the scavenging efficiency remains the same. Riming rate is determined by the terminal velocity of snowflakes, ice crystals and liquid water contents in clouds (Fukuta et al., 1999).

Previously, only the hygroscopicity of BC containing particles is considered in BC scavenging efficiency in models (Wang et al., 2011, and references therein). It is typically assumed that 100% of hydrophilic BC particles are readily incorporated into cloud drops and all hydrophobic BC particles remain in the interstitial air in warm and mixed-phase clouds. This treatment of mixed-phase clouds as liquid phase is likely to overestimate the scavenging rate in mixed-phase clouds. In models that include mixed-phase clouds, assumptions still need to be made about scavenging efficiency. A uniform scavenging efficiency (0.4 or 0.06) for

all mixed-phase clouds has been imposed (Stier et al., 2005; Bourgeois and Bey, 2011), while observations show that scavenging efficiency varies dramatically with temperature and ice mass fraction (Cozic et al., 2007; Henning et al., 2004; Verheggen et al., 2007).

In Experiment D (Table 2), we discriminate WBF-dominated and riming-dominated conditions and parameterize BC scavenging

efficiency under the two conditions separately in mixed-phase clouds (248 K < T < 273 K, Garrett et al., 2010). We assume that riming dominates when temperature is around -10°C (261 K < T < 265 K) and liquid water content is above 1.0 g m$^{-3}$, following Fukuta et al. (1999). The WBF process dominates otherwise. Our parameterization of the effect of the WBF process on BC scavenging efficiency is based on the measurements at Mt. Jungfraujoch (46.4°N, 8°E, 3850 m), an elevated mountainous site far from pollution sources and regularly engulfed in clouds (30% of the time) (Cozic et al., 2007). These characteristics make the

site well suited for investigating continental background aerosols and clouds from a ground based platform. We evaluated the effect of WBF to global BC simulation and tested the sensitivity of the simulation to the switch temperature from warm clouds to mixed-phase clouds and from mixed-phase clouds to ice clouds in a companion study (Qi et al., 2016). In this study, we focus on the effect to BC distribution in the Arctic.

### 3.5 BC concentration in snow

In snow models, such as SNICAR, the initial surface BC$_{snow}$ is defined as the ratio of BC deposition to snow precipitation (Flanner et al., 2007). Here we approximate BC$_{snow}$ using BC deposition flux and snow precipitation rate, following Kopacz et al. (2011) and He et al. (2014a):





$$[BC_{snow}] = \frac{F_{BCdep}}{F_{snow}} = \frac{F_{wet\_dep} + F_{dry\_dep}}{F_{snow}} , \qquad (2)$$

where $F_{BC,dep}$, $F_{wet\_dep}$ and $F_{dry\_dep}$ are total, dry and wet deposition flux of BC and $F_{snow}$ the snow precipitation. The top and bottom snow depth of each sample are provided in the observation dataset (Doherty et al., 2010). We calculate the dates of the snow fall of the top and bottom depth of the snow sample first and then use the average BC deposition fluxes and snow

precipitation between the two dates to estimate the BC snow concentration for the sample. Snow depth is accumulated from the observation date backward until it reaches the top or bottom snow depth of the sample, then the dates are stored. The accumulation rate is estimated as snow precipitation flux (kg m$^{-2}$ s$^{-1}$) over snow density (kg m$^{-3}$). The annual average snow density is 300 kg m$^{-3}$ over the Arctic basin, increasing from 250 kg m$^{-3}$ in September to 320 kg m$^{-3}$ in May with little geographical variation across the Arctic (Warren et al., 1999; Forsström et al., 2013). We use the annual average snow density in

the estimate.

The above estimate of BC$_{snow}$ ignores many processes that may alter the BC snow concentrations, such as wind-redistribution of surface snow, sublimation, and melt water flushing (Doherty et al., 2010, 2013; Wang et al., 2013). Wind-redistribution of surface snow is a sub-grid scale phenomenon. Except for turbulent scale wind direction and strength, small-scale topography also

plays an important role in surface snow redistribution. So this process is really difficult to simulate by global models. Precipitation rate and relative humidity in much of the Arctic are low, so in some areas appreciable (up to 30-50%) surface snow is lost to sublimations (Liston and Sturm, 2004). BC$_{snow}$ at surface can thus be underestimated by our method. We filtered snow samples collected during melting season, so the melt water flushing has little effect to our estimate.

To reduce the biases in comparison of model results and observations, we organize the observations as follows: (1) Observations from March to May in 2007-2009 are used while those from June to August are excluded, because our estimate of BC$_{snow}$ does not resolve snow melting; (2) We exclude observations with obvious dust or local wood-burning contaminations as described in Doherty et al. (2010); (3) We average the observations in the same model grid and snow layer and collected on the same day.

Table 2 summarizes various model simulations in the present study. Experiment A is the standard case. We include gas flaring emissions in Experiment B (Sect. 3.2). Contrasting Experiments B and A thus offer insights to the contribution of gas flaring emissions on BC in the Arctic. Experiment C includes the updated dry deposition velocity (Sect. 3.3). The difference of Experiment B and C denote the effects of updated dry deposition velocity to BC distribution. Experiment D includes temperature-based WBF parameterization (Sect. 3.4). The effects of WBF to BC in the Arctic are shown by the difference of

Experiment C and D. Additional simulations are described where appropriate.

## 4. The effects of gas flares, dry deposition, WBF and precipitation

We discuss the effect of gas flaring emissions, dry deposition, WBF in mixed-phase clouds, and precipitation to BC distribution in the Arctic in this section. The probability density function (PDF) of observed and GEOS-Chem simulated BC$_{snow}$ in the Arctic is approximately lognormal (Fig. 2(a)). The arithmetic mean of observations is 17.4 ng g$^{-1}$, larger than the geometric mean of

12.7 ng g$^{-1}$ and the median of 11.8 ng g$^{-1}$ (see the vertical lines in Fig. 2 and Table 1). The model reproduces the observed distribution, but underestimates BC$_{snow}$ by 40% (Experiment A). By including flaring emissions (Sect. 4.1), updating dry deposition velocity (Sect. 4.2) and including WBF in mixed-phase clouds (Sect. 4.3), the discrepancy is reduced to -10%. Gas



flaring emissions lower the discrepancy from -40% to -20% (Experiment B). The updated dry deposition velocity (Experiment C) makes insignificant changes to $BC_{snow}$ in the Arctic. WBF (Experiment D) further reduces the discrepancy from -20% to -10%. The resulted $BC_{snow}$ in the eight sub-regions agree with observations within a factor of two. In addition, $BC_{air}$ at surface and in the free troposphere is sensitive to the above three processes in the Arctic, particularly during winter and spring (see Sects. 4.1– 4.3).

### 4.1 Gas flaring emissions

Gas flaring emissions increase total BC emissions by 67% (from 0.068 to 0.115 Tg yr$^{-1}$) in the Arctic Circle (60°N and higher latitudes), resulting in a 19% increase of the total BC deposition (from 0.32 to 0.38 Tg yr$^{-1}$). Flaring emissions increase $BC_{snow}$ (by 0.1−8.5 ng g$^{-1}$) in the eight Arctic sub-regions. The higher $BC_{snow}$ leads to significantly reduction in the negative biases (by 20–100%), except in the Arctic Ocean and in Tromsø, where $BC_{snow}$ is already overestimated without flaring emissions (Fig. 3). $BC_{snow}$ in Greenland is not affected by gas flaring emissions. The reason is two-folded: first, snow samples in Greenland are far from the flares in Western Russia; second, the vertical transport of BC from surface to the upper troposphere is suppressed by the stable atmosphere in the Arctic (Stohl, 2006), resulting in negligible effect of flaring emissions to $BC_{snow}$ over Greenland (above 1.5 km).

The largest enhancement of $BC_{snow}$ from flaring emissions is in the Western Extreme North of Russia within the Arctic Circle (by 5.0 ng g$^{-1}$ on average, or, 50%), which reduces model discrepancy substantially across Russia (from -50% to -30%). However, simulated $BC_{snow}$ is now too high by a factor of two near the flares (observed value ~19.3 ng g$^{-1}$). The overestimate is likely because of excessively large flaring emission estimates. Yet $BC_{snow}$ is too low by a factor of two in far fields (observed value ~30.7 ng g$^{-1}$), despite a large increase (by 50%, from 10.5 to 15.5 ng g$^{-1}$) as a result of flaring emissions.

Flaring emissions are assumed to be proportional to flared gas volumes and emission factors. Errors in estimates of flared volumes in Russia is small (within ±5%, Elvidge et al., 2009). Estimates of emission factors, on the other hand, are known to have several orders of magnitude uncertainties (Schwarz et al., 2015; Weyant et al., 2016). Given limited observations of BC emission factors from actual flares, Stohl et al. (2013) derived BC emission factor based upon emission factors of particulate matter from flared gases. They used a BC emission factor of 1.6 g m$^{-3}$, which is more than a factor of three higher than that (0.5 g m$^{-3}$) from a lab experiment on fuel mixtures typical in the oil and gas industry (McEwen et al., 2012). Recent field measurements have suggested an even lower emission factor (0.13±0.36 g m$^{-3}$) from ~30 individual flares in North Dakota, with an upper bound of 0.57 g m$^{-3}$ (Schwarz et al., 2015; Weyant et al., 2016). These studies found that average BC emission factors for individual flares varied by two orders of magnitude, and furthermore, two flares from the same flare stack that were resampled on different days showed different BC emission factors (Weyant et al., 2016). They also pointed out that emission factors are not correlated with ambient temperature, pressure, humidity, flared gas volumes or gas composition. It is thus imperative that extensive measurements of BC emission factors be made in the flare regions.

Yet another source of uncertainty is flare stack height, which is not accounted for in current flaring emission estimates. Typical stack heights vary from 15 to 250 m, sometimes above the nighttime boundary layer eight of 150−300 m in the Arctic (Di Liberto et al., 2012). The stack height affects the ventilation, dispersion, deposition, and long-range transport of the emissions. For example, local deposition of BC may be suppressed and downwind long-range transport enhanced when the stacks elevated BC emissions to the free troposphere (Chen et al., 2009). The lack of proper treatment of flare stack height in the model may





partially explain the aforementioned discrepancies of modeled $BC_{snow}$ (biased high in Western Russian and low in Eastern Russia). Another factor for the underestimate of $BC_{snow}$ in Eastern Russia is likely local sources, such as domestic wood burning in nearby villages and fishing camps, diesel trucks on highway and coal burning in a power plant, that are unaccounted for in the emission inventory (Doherty et al., 2010, Fig. 1). Although we filter out samples with strong local contamination, it is conceivable that local emissions still add to the background $BC_{snow}$ in Eastern Russia.

Jiao et al. (2014) have shown that most AeroCom models underestimated $BC_{snow}$ in Russia and pointed to the flaring emissions as a likely cause. Our results show that even with flaring emissions, which are likely on the high side, $BC_{snow}$ is still too low (by 50%) in Eastern Russia. Therefore, there are likely other factors such as the lack of local emissions in Eastern Russia, weak dry deposition fluxes (Sect. 4.2), and excessively low rate of sublimation of surface snow, that contribute to the large model discrepancy in $BC_{snow}$.

Fig. 4 shows observed and GEOS-Chem simulated daily $BC_{air}$ from January to March at Zeppelin, a site that is closest to the gas flares in the Western Extreme North of Russia. The inclusion of flaring emissions captures some of the large spikes in the observed $BC_{air}$, such as those from late February to March in 2008 and in January 2009. Stohl et al. (2013) found that flaring emissions captured observed large spikes at Zeppelin during a transport event in February 2010 with a high BC/CO ratio, a signature of gas flaring emissions (CAPP, 2007). The inclusion of flaring emissions results in enhanced $BC_{air}$, for instance, in February 2007 and in January 2008, that are not seen in the observations. This is largely from the lack of temporal variation of flaring emissions (Weyant et al., 2016). The temporal variation is, however, difficult to characterize based on the current knowledge of flaring emissions in the Western Extreme North of Russia (Stohl et al., 2013). Flaring emissions also increase $BC_{air}$ during snow season (Sep. to Apr.) (by 16−19 ng m$^{-3}$) at Barrow and Alert, resulting in substantial reductions of discrepancies (from -47% to -15% at Barrow and -67% to -46% at Alert) (Fig. 5). Flaring emissions are transported to the high Arctic within the Arctic dome by efficient circumpolar transport (Stohl, 2006). The effect of flaring emissions at Denali in low Arctic is negligible, because the site is outside of the cold Arctic front (around 65−70°N in Alaska) (Barrie, 1986; Ladd and Gajewski, 2010). The front is a strong barrier for the meridional transport of BC (Stohl, 2006). $BC_{air}$ at Summit (3.22 km a.s.l.), which is mostly in the free troposphere, is not affected by flaring emissions, either. This is because the vertical transport of BC is suppressed by the stable atmosphere during snow season in the Arctic (Stohl, 2006).

## 4.2 Dry deposition velocity

It is known that $v_d$ of aerosol particles over snow and ice surfaces strongly depend on particle size, surface types and meteorological conditions and vary by orders of magnitude (Table 3). We estimate $v_d$ of BC particles as a function of particle properties, aerodynamic resistance and surface types (Sect. 3.3). The results over the Arctic Ocean and Greenland are shown in Table 3, generally within the observed range. At Mt. Changbai, model result of BC $v_d$ (0.09−0.14 cm s$^{-1}$) is an order of magnitude lower than that derived by Wang et al. (2014) (0.16−1.52 cm s$^{-1}$). The resulting dry deposition fluxes are lower than observations by a factor of five. We attribute the large discrepancies to two factors. First, the point measurements were at a mountainous site with complex terrain and micro-meteorological conditions. Neither can be resolved in a global model (He et al., 2014a). Second, the values reported by Wang et al. (2014) were estimated from relative enhancements of surface $BC_{snow}$ between two snow events. These estimates are known to have large uncertainties (a factor of two) from the measured sublimation fluxes and the assumption of snow density (Wang et al., 2014).





Compared to the results of uniform $v_d$ of 0.03 cm s$^{-1}$ over snow and ice, the updated $v_d$ leads to larger dry deposition fluxes, a larger fraction of dry over total deposition, and relatively unchanged total deposition fluxes. Simulated mean BC $v_d$ in the eight Arctic sub-regions (Fig. 1) are 0.03–0.14 cm s$^{-1}$, considerably larger that the uniform value of 0.03 cm s$^{-1}$ over snow and ice (Table 5). Correspondingly, the $v_d$ are 19−195% larger in most sub-regions, with the largest increase in Greenland (by 195%) and

over Russia (by 87%) (Table 5). We find that BC dry deposition flux is more sensitive to $v_d$ in source regions (e.g., Russia) than in remote regions, reflecting the high BC$_{air}$ in the former. A comparable increase in $v_d$ of BC (from 0.03 cm s$^{-1}$ to 0.08 cm s$^{-1}$) in Russia and Alaska results in vastly different increases in BC dry deposition flux (87% in Russia versus 30% in Alaska). As expected, larger dry deposition flux depletes BC$_{air}$ thereby reduces wet deposition flux but offsets the reduction in wet deposition. As a result, both total deposition flux and BC$_{snow}$ remain relatively unchanged (< 5%) in the eight sub-regions, except in

Ny_Ålesund and Tromsø. In these last two regions, the total deposition fluxes are 10−15% smaller. The lower deposition fluxes reflect efficient removal of BC aerosols over source regions. BC in Ny_Ålesund and Tromsø are primarily from Europe and Russia, transported isotropically in cold season (Stohl, 2006; Eleftheriadis et al., 2009). Rapid dry deposition in these source regions results in enhances boundary layer removal hence lower BC loadings in air and a reduced boundary layer outflow (Liu et al., 2011).

The change in the fraction of dry to total deposition has important implications for BC radiative forcing in the Arctic. The fraction increases from 19% (7−33%) to 26% (14−41%), by 14−73%, with the largest increase in Russia (from 23% to 40%) where BC deposition flux and BC$_{snow}$ are the largest in the Arctic (Tables 4 and 5). Typically, BC particles removed by dry deposition are externally mixed with snow particles, while those removed by wet deposition are internally mixed with snow

particles (Flanner et al., 2009, 2012). Internal mixing of BC with snow/ice particles increases the absorption cross-section of BC/snow composites by about a factor of two (Flanner et al., 2012). The enhanced absorption further increases the snow albedo radiative forcing (He et al., 2014b). It is thus conceivable that the larger dry deposition fraction will lead to less internally mixed BC/snow composite and lower snow albedo radiative forcing.

Unlike BC$_{snow}$, BC$_{air}$ is a strong function of dry deposition velocity, particularly during snow season. With updated $v_d$, model results fail to capture the seasonal cycle of BC$_{air}$ with dramatic decreases during snow season (by 20−23 ng m$^{-3}$, 27−68%) at Barrow, Alert, and Zeppelin (Fig. 5). The decreases at Barrow and Alert are a direct result of larger dry deposition in the boundary layer because of substantially larger $v_d$ (0.07 cm s$^{-1}$, Table 5). At Zeppelin (in Ny_Ålesund), where $v_d$ is only marginally higher (17%), the large reduction of BC$_{air}$ (~40%) is largely attributed to the suppressed transport from proximate

source regions in Europe and Russia. This dramatic decrease of BC$_{air}$ in winter with larger $v_d$ and the lack of winter and spring Arctic haze is one of the major reasons of using low $v_d$ in previous studies (Wang et al., 2011; Sharma et al., 2013; Liu et al., 2011). However, this does not justify the use of a low $v_d$ over snow and ice. First, observations have shown very large variations of $v_d$ (Table 3), which suggest that a uniform representation might involve large uncertainties. Second, observations of $v_d$ over snow and ice show very large values in certain region, which is still underestimated by the resistance-in-series method. Third,

besides dry deposition in boundary layer, BC$_{air}$ is affected by a lot of other factors, such as emissions, transport and wet deposition (Sect. 4.3).

### 4.3 WBF in mixed-phase clouds

In mixed-phase clouds, when environmental vapor pressure is between the saturation vapor pressures of water and ice, ice crystals grow at the expense of water drops and release BC particles in cloud water drops back to the interstitial air, now





commonly referred to as the WBF effect (Wegener, 1911; Bergeron, 1935; Findeisen, 1938). Our results show that WBF increases $BC_{snow}$ by 20−80% in the eight Arctic sub-regions, except Canadian sub-Arctic, and increases $BC_{air}$ during snow season by 25−70% (Figs. 2 and 7). Clearly WBF suppresses the scavenging of BC in mixed-phase clouds and consequently enhances poleward transport. We report this in a detailed analysis in a companion study (Qi et al., 2016).

WBF not only increases $BC_{snow}$ in the Arctic but also changes the partition of dry and wet deposition of $BC_{snow}$. Intuitively WBF slows down wet scavenging, thus allowing more BC particles available for dry deposition. Our results show that the fraction of dry to total deposition increases from 26% (12−41%) to 35% (19−59%) on average in the eight Arctic sub-regions, thereby lowering the absorption of solar radiation due to less internally mixed BC-snow composite (Sect. 4.2). In Alaska, Canadian

Arctic and Russia, BC removed by dry deposition increases to more than 50%. However, averaged globally, this fraction increases only slightly (from 19% to 20%), indicating that the fraction in the Arctic is more sensitive to the WBF effect.

The scavenging efficiency of BC, heretofore defined as the fraction of BC incorporated in cloud water drops or ice crystals in mixed-phase clouds, is strongly affected by WBF and as a result varies temporally and spatially in response to varying

temperature (Sect. 3.3). Thus, improved treatment of mixed-phase cloud processes, such as WBF and riming, is essential to improve the simulation of spatial and temporal distribution of BC. BC in Alaska and the Canadian Arctic are most sensitive to the WBF effect in the Arctic. WBF increases $BC_{snow}$ by 55% in Alaska and 43% in the Canadian Arctic and reduces the model discrepancies to within 10% (Table 4 and Fig. 3). $BC_{air}$ at Barrow in Alaska and at Alert in Canadian Arctic are higher by 20−30 ng m$^{-3}$ in winter, reducing the model discrepancies significantly (from -54% to -18% at Barrow and from -72% to -46% at Alert)

and enhancing the seasonal variation (Fig. 5). Similar improvements are also seen at Summit in Greenland, where $BC_{air}$ increases by 12 ng m$^{-3}$ and the model discrepancy lowers significantly (from -48% to 3%). This is consistent with recent observations, which showed that high riming rate was rare (12%) in the North American sector of the Arctic and that WBF dominated in-cloud scavenging in mixed-phase clouds (Fan et al., 2011).

At Zeppelin where snow samples show rimed structures (Hegg et al., 2011), model discrepancy of $BC_{air}$ increases to 63% from -10% with the WBF effect included. Model results do not capture the magnitude of $BC_{air}$ in winter at Barrow, Alert and Zeppelin (Fig. 5). $BC_{air}$ is well simulated at Zeppelin but underestimated at Barrow and Alert in Experiment A. $BC_{air}$ is well simulated at Barrow and Alert but overestimated at Zeppelin in Experiment D (Fig. 5) – similar results were shown in Sharma et al. (2013). Such apparent discrepancy can be partly attributed to the fact that models do not properly distinguish WBF-dominated in-cloud

scavenging at Barrow (Fan et al., 2011) and riming-dominated scavenging at Zeppelin (Hegg et al., 2011). Here we separate WBF- and riming-dominated conditions based on temperature and liquid water content (LWC) (Sect. 3.3, and Fukuta et al., 1999) in Experiment D. However, model results still fail to capture the difference among the three sites. There are a number of reasons. First, LWC from GEOS-5 biased high compared to CloudSat observations (Barahona et al., 2014). In addition, the spatial distribution of LWC from GEOS-5 also has large discrepancy (Li et al, 2012; Barahona et al., 2014). Second, this separation is

based on a laboratory experiment, while conditions in the real atmosphere are much more complex. Therefore, more field measurements are required to better separate the two conditions and better parameterize BC scavenging efficiency.

Our results show that WBF exaggerates the positive bias of $BC_{air}$ in summer and delays the transition from the late-spring haze to the clean summer boundary layer (Experiment D). Previous studies found that the dominant process controlling low summertime

aerosol at Barrow is the onset of local wet scavenging by warmer clouds (Garrett et al., 2010, 2011). WBF suppresses





scavenging in mixed-phase clouds and thus slows down the onset of strong scavenging by warmer clouds during the transition from winter to summer. However, the strong scavenging of warm drizzling clouds in late spring and summer boundary (Browse et al., 2012), which enhances the winter-summer transition, is not considered in the present study. At high latitudes in summer, the relatively high humidity allows the formation of low stratocumulus cloud decks in the boundary layer and lower troposphere

(Browse et al., 2012). These low clouds and fogs produce frequent drizzle and can be present for as much as 90% of the time and remove aerosols effectively (Browse et al., 2012).

## 4.4 Precipitation

We compute $BC_{snow}$ as the ratio of BC deposition flux to precipitation rate (Sect. 3.5). It has been pointed out that this estimate is very sensitive to uncertainties in precipitation (He et al., 2014a). Climatological precipitation across the Arctic is 14.3 g cm$^{-2}$ yr$^{-1}$

for 1965–89 (Overland and Turet, 1994) and is 16.3 g cm$^{-2}$ yr$^{-1}$ for 1971–91 (Serreze et al., 1995) as constrained from observed hydrologic budget (Warren et al., 1999). The annual precipitation, averaged for 2007–09, is 15.5 g cm$^{-2}$ yr$^{-1}$ in GEOS-5, within the range of the observations. There are considerable uncertainties, spatially and temporally, in precipitation in the Arctic (Warren et al., 1999; Serreze et al., 2000). Fig. 6 compares monthly precipitation from the Global Precipitation Climatology Project (GPCP, Huffman et al., 2001), NOAA Climate Prediction Center Merged Analysis of Precipitation (CMAP, Xie and

Arkin, 1997), and GEOS-5. The discrepancies can be as large as a factor of 10 and the seasonal cycles are largely out of phase between the three datasets. Specifically, GPCP precipitation is much stronger than CMAP, particularly during summer. GEOS-5 precipitation is within the range of GPCP and CMAP data. The exception is Greenland, Ny_Ålesund, and Tromsø, where GEOS-5 precipitation is substantially (a factor of 2–10) larger than GPCP and CMAP data during the snow season. Snow in the Arctic is difficult to constrain for two reasons. First, accurate measurements of snowfall in the Arctic have proven nearly impossible,

because snow gauges strongly under-catch snowfall (by 55−75%) depending on the gauge type and wind condition (Liston and Sturm, 2004). Second, although it is possible to correct for the low bias, a more fundamental problem is the sparse observational network (Serreze et al., 2000). Previous studies showed that 10–40 stations are required to provide accurate monthly mean precipitation estimates for 2.5° latitude–longitude bands (WCRP, 1997). To that end, the number of station in the Arctic is vastly inadequate (Serreze et al., 2000).

To probe the sensitivity of BC deposition and $BC_{snow}$ to precipitation, we conduct two additional model simulations, where we halve and double precipitation in the Arctic, with other processes configured as in Experiment D. We find that, in GEOS-5, during the snow season, nearly all precipitation is in the form of snow in the Arctic. Halving precipitation leads to increases in $BC_{snow}$ by 15−136%, with largest enhancements in Greenland (136%) and Ny-Ålesund (92%) (Fig. 7). With precipitation halved,

it takes a longer accumulation time for a given snow depth, which results in larger dry deposition (up to 153% increases). As such, the ratio of BC dry deposition to snow precipitation increases as well. On the other hand, the ratio of BC wet deposition to snow precipitation, determined mainly by in-cloud scavenging of BC, remains largely unchanged. Overall, $BC_{snow}$ increases with halved precipitation. It is conceivable that doubled precipitation has the opposite effect. Indeed, $BC_{snow}$ decreases by 14−43% in the eight Arctic sub-regions. In addition, dry deposition decreases by 35−62% and the fraction of dry to total deposition

decreases by 23−43%. Although $BC_{snow}$ as computed here is sensitive to precipitation, the resulting medians of $BC_{snow}$ in the eight sub-regions are in agreement with observations within a factor of two, except over Greenland (a factor of five too high) and Tromsø (a factor of three too high). Further analysis of the results at Greenland and Tromsø is in Sect. 4.5. The strong sensitivity of $BC_{snow}$ calls for better constraining of precipitation in the Arctic.



In contrast, annual BC burden and deposition are much less sensitive to precipitation. Halving Arctic precipitation increases annual BC burden by 12% and decreases annual BC deposition by 16% in the Arctic. This is because less precipitation removes less BC particles. BC lifetime in the Arctic, as determined by the BC burden and deposition, increases by 27%. When precipitation is doubled, annual BC burden decreases by 14%, while BC deposition increases by 8%, resulting in a 23% reduction of BC lifetime in the Arctic.

$BC_{air}$ is more sensitive to precipitation at Barrow, Alert and Zeppelin than at Denali and Summit (Fig. 8). When precipitation is halved, annual $BC_{air}$ increases by 20−70% at Alert, by 10−40% at Barrow and Zeppelin, and by 1−20% at Denali and Summit. When precipitation is doubled, annual $BC_{air}$ decreases by 20−50% at Alert, by 10−40% at Barrow and Zeppelin, and by 2−20% at Denali and Summit. Additionally, $BC_{air}$ is more sensitive to precipitation in summer than in winter. This is because the summer clean boundary layer in the Arctic is controlled by strong local scavenging (Garrett et al., 2010, 2011; Browse et al., 2012).

**4.5 BC in snow in Greenland, Tromsø and Canadian sub-Arctic**

$BC_{snow}$ is associated with much larger uncertainties over short (hence shallower snow depth) than longer (hence larger snow depth) time periods. Because snow samples over Greenland were collected at the very surface (~0 cm), the computed $BC_{snow}$ thus represents BC deposition only through the duration of a day for direct comparisons. The short time duration thus largely explains the larger uncertainties in the estimated $BC_{snow}$. In Tromsø, observed $BC_{snow}$ were considerably lower (19.1 ng g$^{-1}$) from samples collected over a clean mountain plateau upwind of town Tromsø (Doherty et al., 2010) and much higher (53.3 ng g$^{-1}$) from samples collected in town (Forsström et al. 2013). We use the former for comparisons. Thus, the factor of two overestimate of $BC_{snow}$ in this region is because that GEOS-Chem does not resolve sub-grid variability.

In the Canadian sub-Arctic, $BC_{snow}$ is underestimated by 50% with all the improvements discussed above (Experiment D). This large low bias is mainly from the low $BC_{snow}$ in the subsurface samples (1−20 cm, 11.7 ng g$^{-1}$, ~60% of all samples), accumulated through the snow season. $BC_{snow}$ in this region increases by 33% from flaring emissions and by 43% from halving precipitation. Yet the resulting $BC_{snow}$ is still 25% lower than observations (12.8 ng g$^{-1}$). However, GEOS-5 precipitation is at the lower end among the three precipitation datasets (Fig. 6). The large discrepancy in $BC_{snow}$ warrants further studies.

**5. Discussions**

Global BC emissions in this study are within the range of previous studies (Table 6). Gas flares are a rather small fraction (3%) of the global BC emissions but a dominant BC source in the Arctic – it is 41% of the total BC emissions in the Arctic. The resulting total BC emissions (0.115 Tg yr$^{-1}$) and deposition (0.38 Tg yr$^{-1}$) in the Arctic exceed the higher end of those used in previous studies (emissions: 0.037−0.077 Tg yr$^{-1}$, depositions: 0.13–0.34 Tg yr$^{-1}$) (Table 6). BC deposition is strongly sensitive to precipitation. Doubling (halving) precipitation in the Arctic increases (decreases) BC deposition by 8% (18%). Total BC emissions in the Arctic is a factor of two to five lower than the total BC deposition, suggesting that a large fraction of BC is from long-range transport.

Estimates of BC loading in the Arctic are considerably uncertain. The estimates from AeroCom range from 0.02 to 0.34 mg m$^{-2}$, much larger than the spread of emissions (a factor of two) and depositions (a factor of three) (Table 6). We find that BC loading





in the Arctic is more sensitive to wet scavenging efficiency than to emissions and $v_d$. BC loading in the Arctic increases by 13% from flaring emissions, which represents a ~70% enhancement to previous emission estimates, and by 7% from updated $v_d$, which in some cases are a factor of two to three larger. WBF reduces BC scavenging efficiency in mixed-phase clouds by 20−80% and increases annual BC loading by 70% in the Arctic. The resulting BC loading of 0.43 mg m$^{-2}$ exceeds the high end of the AeroCom models (0.02−0.34 mg m$^{-2}$). In addition, Arctic BC loading increases by 12% when precipitation is halved and decreases by14% when precipitation is doubled. We also find that BC lifetime in the Arctic is very sensitive to BC scavenging. As an example, WBF increases the BC lifetime from ~9 days to ~16 days.

The large variation of the BC loadings from the AeroCom models can largely be attributed to different treatments of BC scavenging efficiency in mixed-phase clouds and in ice clouds. Bourgeois and Bey (2011) reduced the scavenging efficiency in mixed-phase clouds from 0.10−0.75 to a uniform value of 0.06 in the ECHAM5-HAMMOZ model (Pozzoli et al., 2008) and found that the resulting BC$_{air}$ in the Arctic increased by up to a factor of ten and were in improved agreement with aircraft observations. In addition, their model results of BC burden in the Arctic were five times higher. We note here that a scavenging efficiency of 0.06 is on the low end of observed values in mixed-phase clouds (Cozic et al., 2007; Verheggen et al., 2007), which leads to a considerably larger WBF effect. Liu et al. (2011) found that lowering BC scavenging efficiency in ice clouds (from 0.2 to 0.01) in the AM3 model (Anderson et al., 2004) dramatically enhanced BC transport to the Arctic (nearly 10 times higher) and improved model comparison with aircraft observations. Browse et al. (2012) suppressed the scavenging of soluble BC in ice clouds in the GLOMAP model (Mann et al., 2010) and found that the resulting BC$_{air}$ in the Arctic were six times higher. Better characterization of scavenging efficiency in all could types globally is thus critical for accurately reproducing BC distribution and the associated climatic effects in the Arctic.

Flaring emissions improve the agreement of BC$_{snow}$ with observations significantly, with a 50% reduction to the negative bias of modeled BC$_{snow}$ across the Arctic and a substantially stronger correlation (0.15 to 0.24) between simulated and observed BC$_{snow}$ in the region (Table 6). WBF further reduces the average bias across the Arctic by 70%. BC$_{snow}$ is also sensitive to precipitation in the Arctic. Doubling (halving) precipitation introduces a much larger positive (negative) bias, similar as the magnitude of the overall effects of flaring emissions and the WBF effect.

There is a large divergence (a factor of 5−6) in BC$_{snow}$ in the Arctic from the AeroCom models (Jiao et al., 2014). BC$_{snow}$ in the Arctic are biased low in nearly all the AeroCom Phase I models (with the exception of ULAQ) but biased high in the AeroCom Phase II models (except for CAM4-Oslo and CAM5.1) (Table 6). The seasonal cycle of BC deposition varies significantly among the models (Jiao et al., 2014). It is likely that monthly mean BC deposition fields used in AeroComp introduce biases of 1.5−2.5 because of the mismatch between the temporally smoothed deposition fields and episodic snow events (Doherty et al., 2014). Additionally, meteorological fields including precipitation in the Arctic are known to be poorly constrained due to the scarcity of observations in the region (Sect. 4.4). Overall, modeled BC$_{snow}$ is poorly correlated with observations ($r = 0.12−0.24$) for all AeroCom models and GEOS-Chem.

## 6. Summary and conclusions

This study sought to understand the capability of GEOS-Chem in simulating BC distribution in the Arctic and the controlling factors of BC distribution. We evaluated the model simulation against BC$_{snow}$ measurements across the Arctic and in-situ



measurements of surface $BC_{air}$ at Denali in lower Arctic, Barrow, Alert and Zeppelin in higher Arctic, and Summit in the free troposphere. We also examined the role of gas flaring emissions, dry deposition velocity, the WBF effect, and precipitation on BC distribution in the Arctic.

We estimated the date of snow accumulation to the top and bottom depth of a snow sample and $BC_{snow}$ was estimated as the ratio of mean BC deposition flux to precipitation flux between the two dates. The model underestimated $BC_{snow}$ in the Arctic by ~40%. Flaring emissions increased $BC_{snow}$ by $0.1−8.5$ ng g$^{-1}$ in the eight Arctic sub-regions and decreased the discrepancy to -20% on average. Among the eight Arctic sub-regions, $BC_{snow}$ over Greenland was not affected by flaring emissions, because the stable atmosphere during snow season in the Arctic strongly suppressed the vertical transport of BC to the upper troposphere (Stohl,

2006). With flaring emissions, $BC_{snow}$ in Western Russia biased high by a factor of two, while $BC_{snow}$ in Eastern Russia biased low by a factor of two. This discrepancy is likely due to excessive emission factors or missing of flare stack height. We call for extensive measurements of flaring emission factors in the Arctic.

     Our analysis suggested that resistance-in-series method captures the large variations of dry deposition velocity over snow and ice
surfaces from observations. The updated $v_d$ (increases from $0.03−0.12$ cm s$^{-1}$ to $0.03−0.24$ cm s$^{-1}$ in the eight sub-regions) did not affect $BC_{snow}$, because higher dry deposition flux was compensated by lower wet deposition flux. The fraction of dry to total deposition increased from 15% to 26%. Thus, the radiative forcing induced by BC deposition in snow could decrease because of fewer internally mixed BC/snow composite. BC deposition in Ny_Ålesund decreased by ~10% due to the reduced boundary layer transport from the proximate BC sources in Russia and Tromsø with updated $v_d$, which strongly increased the boundary
layer deposition of BC in the two source regions. WBF enhanced $BC_{snow}$ by $0.3−5.6$ ng g$^{-1}$ and further decreased the model discrepancy to -10% for the whole Arctic. In addition, WBF increased the fraction of dry to total deposition from 26% to 35%. $BC_{snow}$ in Alaska and Canadian Arctic was most sensitive to the WBF effect, which increased $BC_{snow}$ in the two regions by ~50% and reduced the discrepancy to within 10%. With all above improvements, median $BC_{snow}$ in the Arctic agreed with observations within 10%. In the eight sub-regions, $BC_{snow}$ agreed with observations within a factor of two. In addition, $BC_{snow}$ was sensitive to
precipitation as well. Halving (doubling) precipitation in the Arctic increased (decreased) $BC_{snow}$ by 72-92% (15-36%).

     $BC_{air}$ in the Arctic was sensitive to flaring emissions, dry deposition velocity and the WBF effect in mixed-phase clouds. Gas flaring emissions increased $BC_{air}$ in winter and spring by about ~20 ng m$^{-3}$ in the lower troposphere (Barrow, Alert and Zeppelin), while $BC_{air}$ in the free troposphere (Summit) was not affected by flaring emissions. $BC_{air}$ decreased by about ~20 ng m$^{-3}$ during
snow season in high Arctic. WBF increased $BC_{air}$ by ~20 ng m$^{-3}$ during snow season and offseted the decrease cause by increasing $v_d$. However, WBF also increased $BC_{air}$ during the transition period from winter haze to clean summer boundary layer and slows down the transition. This can be attributed to the missing of scavenging by boundary layer drizzle in late spring and summer in the Arctic (Browse et al., 2012). In addition, our scheme with the WBF effect did not capture the magnitude of $BC_{air}$ at Barrow, Alert and Zeppelin, because it did not resolve the WBF-dominated (Barrow and Alert) and riming-dominated in-cloud
scavenging (Zeppelin) properly.

     BC loading and lifetime in the Arctic was very sensitive to the WBF effect. WBF reduced scavenging efficiency by up to 80% globally and increased BC loading in the Arctic from 0.25 mg m$^{-2}$ to 0.43 mg m$^{-2}$ and increased BC lifetime in the Arctic from ~9 days to ~16 days. However, the parameterization of the WBF effect in this study was based on observations at one single site



in Europe in mid-latitudes, we need more observations of BC scavenging efficiency in the Arctic and in other mid-latitude regions to understand the BC-cloud interaction and constrain BC distribution.

**Acknowledgements**

This study was funded by NASA grant NNX14AF11G from the Atmospheric Chemistry Modeling and Analysis Program
5 (ACMAP). The authors thank Y. Kondo, J. P. Schwarz, S. G. Warren, and H. Liu for helpful discussions.

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



**Table 1: Measurements of BC in surface air in the Arctic.**

| Station | Temporal frequency | Data availability | References |
|---|---|---|---|
| **Denali** (63.7°N, 149.0°W, 0.66 km) | 24-h average every 3 days | 91% | Malm *et al.* (1994) |
| **Barrow** (71.3°N, 156.6°W, 0.01 km) | 1 h | 46% | Bodhaine (1989) |
| **Alert** (82.3°N, 62.3°W, 0.21 km) | 1 h | 84% | Sharma *et al.* (2004) |
| **Zeppelin** (79°N, 12°E, 0.47 km) | 30 min | 79% | Eleftheriadis *et al.* (2009) |
| **Summit** (72.6°N, 38.5°W, 3.22 km) | 5 min | 95% | Delene and Ogren (2002) |





**Table 2: GEOS-Chem simulations of BC in the Arctic.**

| Experiments | | A | B | C | D |
|---|---|---|---|---|---|
| **Anthropogenic emissions** | **Arctic** | Bond *et al.* (2007) | colspan | colspan | Bond *et al.* (2007) and flaring emissions from Stohl *et al.* (2013) |
| | **Asia** | Zhang *et al.* (2009) | | | |
| | **Rest of world** | Bond *et al.* (2007) | | | |
| **Biomass burning** | | GFEDv3 (van der Werf *et al.*, 2010), with updates from Randerson *et al.* (2012) | | | |
| **BC aging** | | e-folding time 1.15 days | | | |
| **Deposition** | **Dry** | 0.03 cm s$^{-1}$ over snow/ice and resistance-in-series over other surfaces (Wang *et al.*, 2011) | | Resistance-in-series over all surfaces (Wesely, 1989; Zhang *et al.*, 2001) | |
| | **Wet** | Liu *et al.* (2001) with updates from Wang *et al.* (2011) | | | |
| | | Riming: scavenging efficiency for hydrophilic BC is 100% in warm and mixed phase clouds | | Account for both riming and WBF in mixed phase clouds (Fukuta *et al.*, 1999; Verheggen *et al.*, 2007; Cozic *et al.*, 2007) | |





Table 3: Observed and simulated dry deposition velocity ($v_d$) using resistance-in-series method over snow and ice.

| Region | Sample | Observed $v_d$ (cm s$^{-1}$) | Simulated $v_d$ (cm s$^{-1}$) | References | Particle diameter |
|---|---|---|---|---|---|
| Arctic Ocean | Open water leads, ice ridges, snow and ice surfaces | 0.027-0.068[a] | 0.006-0.070[a] | Held *et al.* (2011) | < 50 nm |
| Arctic Ocean | Open sea | 0.19[b] | | Nilsson and Rannik (2001) | |
| Arctic Ocean | Frozen, partly snow-covered ice | 0.03[b] | 0.013-0.22[b] | Nilsson and Rannik (2001) | Mostly ultrafine and Aitken mode |
| Arctic Ocean | Summer lead | 0.034[b] | | Nilsson and Rannik (2001) | |
| Arctic Ocean | Freeze-up lead | 0.091[b] | | Nilsson and Rannik (2001) | |
| Greenland | Snow (sulfate) | 0.023-0.062[c] | 0.007-0.16[c] | Bergin *et al.* (1995) | < 10 μm |
| Greenland | Snow (sulfate) | 0.01-0.18[d] | 0.007-0.20[d] | Hillamo *et al.* (1993) | 0.6 μm |
| Greenland | Snow | 0.2-0.7 | | Hillamo *et al.* (1993) | 2 μm |
| Antarctic | Snow grass | 0.02-0.1 | | Wesely *et al.* (1979) | 0.05-1.0 μm |
| Antarctic | Smooth snow surface | 0.33 (0.08-1.89) | | Grönlund *et al.* (2002) | 14 nm |
| Antarctic | Rocky surface interrupted by snow | 0.8 (0.2-2.4) | | Grönlund *et al.* (2002) | 42 nm |
| Norway | Snow | 0.06-0.38 | | Dovland and Elliassen (1976) | |
| Pennsylvania | Snow covered farm land in December | 0.034±0.014 | | Duan *et al.* (1988) | 0.15-0.3 μm |
| Mt. Changbai | Snow covered mountain (BC) | 0.16-1.52[e] | 0.09-0.14[e] | Wang *et al.* (2014) | |

[a] This range of measurements are medians of dry deposition velocities derived from aerosol number fluxes measured by an eddy covariance system over different surface types (open water leads, ice ridges, snow and ice surfaces) in the Arctic Ocean between 2°−10° W longitude and 87°−87.5° N latitude in late August 2008 (Held et al., 2011). The simulated dry deposition velocities are sampled at the same region during the same time period as observations for BC particles.

[b] Observations are medians of dry deposition velocities derived from aerosol number fluxes measured by an eddy covariance system over different surface types in late July and early August in 1996 in the Arctic Ocean for ultra fine and Aitken mode aerosol particles (Nilsson and Rannik, 2001). Simulations are sampled in the same region during the same months as observations in 2008 for BC particles.

[c] Sulpate dry deposition velocities were derived based on particle mass using surrogate surfaces and impactor data at site Summit, Greenland in July 1993 (Bergin et al., 1995). Simulations are sampled at the same site during July 2008 for BC particles.

[d] Sulpate dry deposition velocities were derived based on particle mass from Cascade impactor at Dye 3 on the south-central Greenland Ice Sheet in March 1989 (Hillamo et al., 1993). Simulations are sampled at the same site during March 2008 for BC particles.

e The dry deposition velocities specific to BC particles were derived from measured surface enhancement of BC in snow between two snow events at Changbai Mountain in Northern China in winter (December, January, and February) in 2009-2012 (Wang et al., 2014). Simulations are sampled at the same site during the same time period for BC particles.



**Table 4: Observed and GEOS-Chem simulated BC concentration in snow in the Arctic (ng g$^{-1}$, see Fig. 1).**

| | | | Arctic | Alaska | Arctic Ocean | Canadian Sub-Arctic | Canadian Arctic | Greenland | Ny_Ålesund | Russia | Tromsø |
|---|---|---|---|---|---|---|---|---|---|---|---|
| | **Sample size** | | 334 | 3 | 23 | 34 | 86 | 8 | 39 | 118 | 23 |
| **Arithmetic mean** | **Obs.** | | 19.8 | 12.4 | 8.0 | 14.8 | 8.8 | 3.2 | 13.7 | 28.3 | 19.3 |
| | Experiment | **A** | 10.9 (0.6[+]) | 6.0 (0.5) | 8.5 (1.1) | 7.7 (0.5) | 5.7 (0.7) | 3.6 (1.1) | 10.9 (0.8) | 12.3 (0.4) | 35.6 (1.8) |
| | | **B** | 15.0 (0.8) | 7.7 (0.6) | 10.8 (1.4) | 9.3 (0.6) | 6.7 (0.8) | 3.6 (1.1) | 14.9 (1.1) | 19.6 (0.7) | 41.8 (2.2) |
| | | **C** | 15.1 (0.8) | 8.0 (0.6) | 10.3 (1.3) | 9.1 (0.6) | 7.0 (0.8) | 4.3 (1.3) | 12.8 (0.9) | 20.7 (0.7) | 38.4 (2.0) |
| | | **D** | 16.0 (0.8) | 12.2 (1.0) | 12.4 (1.6) | 8.5 (0.6) | 8.8 (1.0) | 5.1 (1.6) | 14.9 (1.1) | 19.4 (0.7) | 45.8 (2.4) |
| **Geometric Mean** | **Obs.** | | 12.9 | 11.4 | 6.8 | 13.2 | 8.2 | 2.7 | 11.2 | 21.2 | 18.8 |
| | Experiment | **A** | 7.6 (0.6) | 5.9 (0.5) | 7.3 (1.1) | 5.9 (0.5) | 4.9 (0.6) | 2.3 (0.9) | 8.4 (0.8) | 9.3 (0.4) | 28.3 (1.5) |
| | | **B** | 10.4 (0.8) | 7.6 (0.7) | 9.6 (1.4) | 7.6 (0.6) | 6.1 (0.7) | 2.4 (0.9) | 11.4 (1.0) | 14.3 (0.7) | 35.1 (1.9) |
| | | **C** | 10.1 (0.8) | 7.9 (0.7) | 9.3 (1.4) | 7.3 (0.6) | 6.3 (0.8) | 2.8 (1.0) | 9.7 (0.9) | 13.9 (0.7) | 31.6 (1.7) |
| | | **D** | 11.5 (0.9) | 11.6 (1.0) | 11.6 (1.7) | 7.6 (0.6) | 8.1 (1.0) | 3.8 (1.4) | 11.9 (1.0) | 14.2 (0.7) | 37.2 (2.0) |
| **Median** | **Obs.** | | 11.8 | 11.0 | 7.6 | 12.8 | 8.9 | 2.5 | 11.9 | 22.1 | 19.1 |
| | Experiment | **A** | 6.9 (0.6) | 6.3 (0.6) | 6.4 (0.8) | 5.5 (0.4) | 4.1 (0.5) | 2.3 (0.9) | 8.4 (0.7) | 10.8 (0.5) | 25.2 (1.3) |
| | | **B** | 9.5 (0.8) | 7.6 (0.7) | 7.7 (1.0) | 7.3 (0.6) | 5.7 (0.6) | 2.3 (0.9) | 11.1 (0.9) | 16.1 (0.7) | 33.7 (1.8) |
| | | **C** | 8.7 (0.7) | 7.8 (0.7) | 8.5 (1.1) | 7.3 (0.6) | 6.0 (0.7) | 3.2 (1.3) | 9.2 (0.8) | 16.1 (0.7) | 29.2 (1.5) |
| | | **D** | 11.0 (0.9) | 12.1 (1.1) | 10.9 (1.4) | 6.8 (0.5) | 8.6 (1.0) | 5.7 (2.3) | 11.3 (1.0) | 16.9 (0.8) | 38.2 (2.0) |

[+]Ratio of model to observation





**Table 5: GEOS-Chem simulated BC dry deposition velocity (cm s$^{-1}$), dry deposition flux (ng m$^{-2}$ d$^{-1}$) and fraction of dry to total deposition (%) in the Arctic.**

| Region | Dry deposition velocity (cm s$^{-1}$) | | Dry deposition flux (ng m$^{-2}$ d$^{-1}$) | | | Total deposition flux (ng m$^{-2}$ d$^{-1}$) | | | Dry deposition fraction (%) | | |
|---|---|---|---|---|---|---|---|---|---|---|---|
| | Exp. B | Exps. C & D | Exp. B | Exp. C | Exp. D | Exp. B | Exp. C | Exp. D | Exp. B | Exp. C | Exp. D |
| Alaska | 0.03 | 0.08 | 787 | 1018 | 1906 | 2393 | 2469 | 3665 | 33 | 41 | 52 |
| Arctic Ocean | 0.03 | 0.07 | 662 | 789 | 1520 | 4480 | 4227 | 4733 | 15 | 19 | 32 |
| Canadian sub-Arctic | 0.04 | 0.08 | 841 | 1192 | 2297 | 5669 | 5596 | 5013 | 15 | 21 | 46 |
| Canadian Arctic | 0.03 | 0.07 | 661 | 988 | 1948 | 3194 | 3289 | 3343 | 20 | 30 | 58 |
| Greenland | 0.03 | 0.10 | 262 | 772 | 1804 | 3887 | 4245 | 4481 | 7 | 18 | 40 |
| Ny_Ålesund | 0.12 | 0.14 | 2654 | 2322 | 4861 | 19528 | 16713 | 19536 | 14 | 14 | 25 |
| Russia | 0.03 | 0.08 | 3092 | 5782 | 7288 | 13647 | 14465 | 12336 | 23 | 40 | 59 |
| Tromsø | 0.12 | 0.13 | 5826 | 5110 | 9339 | 46382 | 42085 | 49598 | 13 | 12 | 19 |



**Table 6: Model simulations of BC in the Arctic (60°N to 90°N).**

| | Model | Global Emission[b] (Tg yr⁻¹) | Arctic Emission[b] (Tg yr⁻¹) | Arctic Deposition[b] (Tg yr⁻¹) | Arctic Loading[c] (mg m⁻²) | Arctic Lifetime[d] (d) | $BC_{snow}$ Bias[e] (ng g⁻¹) | $BC_{snow}$ $r$ [e] | Year of deposition field[b] |
|---|---|---|---|---|---|---|---|---|---|
| **GEOS-Chem[a]** | **Experiment A** | 8.3 | 0.068 | 0.32 | 0.24 | 9.9 | - 5.3 | 0.15* | 2006-2009 |
| | **Experiment B** | 8.5 | 0.115 | 0.38 | 0.27 | 9.5 | - 2.5 | 0.24* | 2006-2009 |
| | **Experiment C** | 8.5 | 0.115 | 0.37 | 0.25 | 9.2 | - 2.9 | 0.23* | 2006-2009 |
| | **Experiment D** | 8.5 | 0.115 | 0.37 | 0.43 | 16.3 | - 0.8 | 0.21* | 2006-2009 |
| | **Exp. D_ 50% precip.** | 8.5 | 0.115 | 0.31 | 0.48 | 20.7 | +5.8 | 0.22* | 2006-2009 |
| | **Exp. D_200% precip.** | 8.5 | 0.115 | 0.40 | 0.37 | 12.6 | -4.4 | 0.20* | 2006-2009 |
| | **AeroCom Phase I[f]** | 7.8 | 0.069 | 0.11-0.22 | - | - | -13.2~-0.5[g] | 0.11-0.28 | - |
| **AeroCom Phase II** | **HADGEM2** | 6.6 | 0.063 | 0.34 | 0.34 | 22.6 | + 18.7 | 0.18* | 2006-2008 |
| | **GOCART** | 10.3 | 0.058 | 0.29 | 0.14 | 16.0 | + 7.3 | 0.04 | 2006 |
| | **OsloCTM2** | 7.8 | 0.068 | 0.28 | 0.07 | 6.9 | + 21.4 | 0.10* | 2006 |
| | **GISS-modelE** | 7.6 | 0.077 | 0.22 | 0.16 | 11.6 | + 7.8 | 0.21* | 2004-2008 |
| | **SPRINTARS** | 8.1 | 0.037 | 0.22 | 0.08 | 6.9 | + 5.3 | 0.06 | 2006 |
| | **CAM4-Oslo** | 10.6 | 0.056 | 0.21 | 0.20 | 22.7 | - 0.2 | 0.12* | Present-day |
| | **GMI** | 7.8 | 0.059 | 0.20 | 0.08 | 7.7 | + 1.9 | 0.10* | 2006 |
| | **IMPACT** | 10.6 | 0.039 | 0.16 | 0.05 | - | + 3.8 | 0.18* | Present-day |
| | **CAM5.1** | 7.8 | 0.056 | 0.13 | 0.02 | - | - 13.0 | 0.23* | 2006 |

[a] This study
[b] AeroCom model results are from Jiao *et al.* (2014).
[c] AeroCom models simulated Arctic Burdens are for year 2000 using only anthropogenic emissions from Samset *et al.* (2013)
[d] Lifetime is approximated by dividing the annual Arctic BC column burden by the annual Arctic deposition flux.
[e] BC snow concentrations were calculated using CLM4 and CICE4 models with monthly deposition field from AeroCom models (Jiao *et al.*, 2014).
[f] Paticipating models are DlR, GISS, LOA, LSCE, MATCH, MPI-HAM, TM5, UIO-CTM, UIO-GCM,UIO-GCM-V2, ULAQ, UMI, CAM-Oslo (Jiao *et al.*, 2014)
[g] This range is for the AeroCom Phase I models except for ULAQ, which is the only one produce a positive bias of +10.7 ng g⁻¹.
*The regression is significant at α=0.05





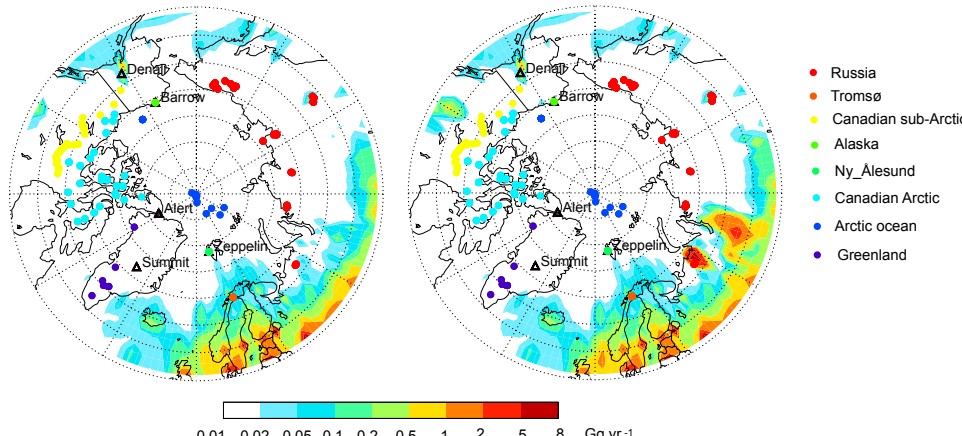

**Figure 1: Annual BC emissions (Gg yr$^{-1}$) in the Arctic in Experiment A (left panel) and Experiments B, C and D (right panel). Also shown are in-situ BC measurement stations (open triangles) and snow sample locations (solid circles). The eight sub-regions of the Arctic as defined in Doherty *et al.* (2010) are color-coded. See text for details.**



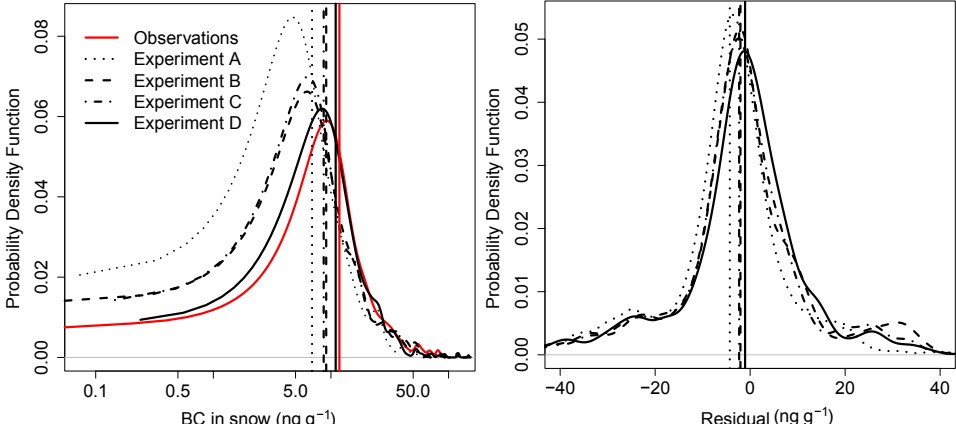

**Figure 2: Probability density function of observed (solid red) and GEOS-Chem simulated (black curves: dotted−Experiment A; dashed−Experiment B; dash dotted−Experiment C; solid−Experiment D, see Table 2 and text for details) BC concentration in snow (ng g⁻¹) in the Arctic (left panel), medians (vertical lines, left panel), residual errors (model−observation, right panel) and mean residual errors (vertical lines, right panel).**





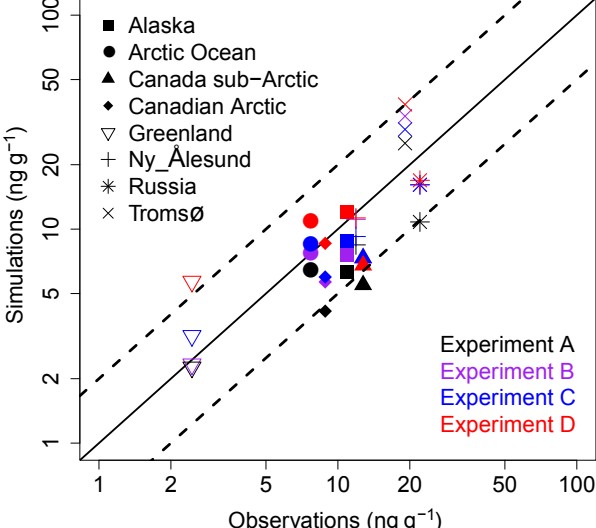

**Figure 3: Observed and GEOS-Chem simulated median BC concentration in snow (ng g⁻¹) in the eight sub-regions in the Arctic (see Fig. 1). Solid line is 1:1 ratio line and dashed lines are 1:2 (or 2:1).**





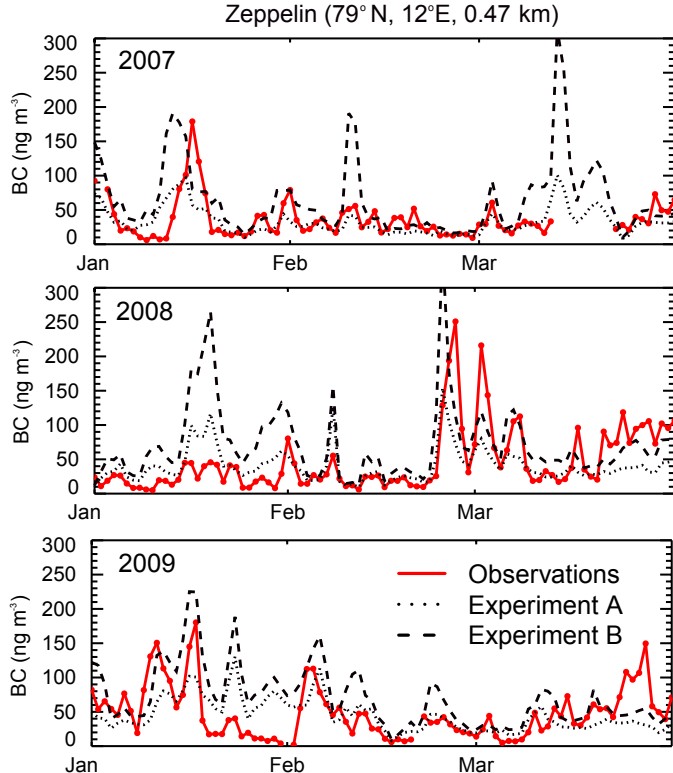

**Figure 4: Observed (red solid) and GEOS-Chem simulated (dotted−Exp. A, dashed−Exp. B, see Table 2 and text for details) daily BC concentrations in air (ng m⁻³) at Zeppelin from January–March in 2007−09.**





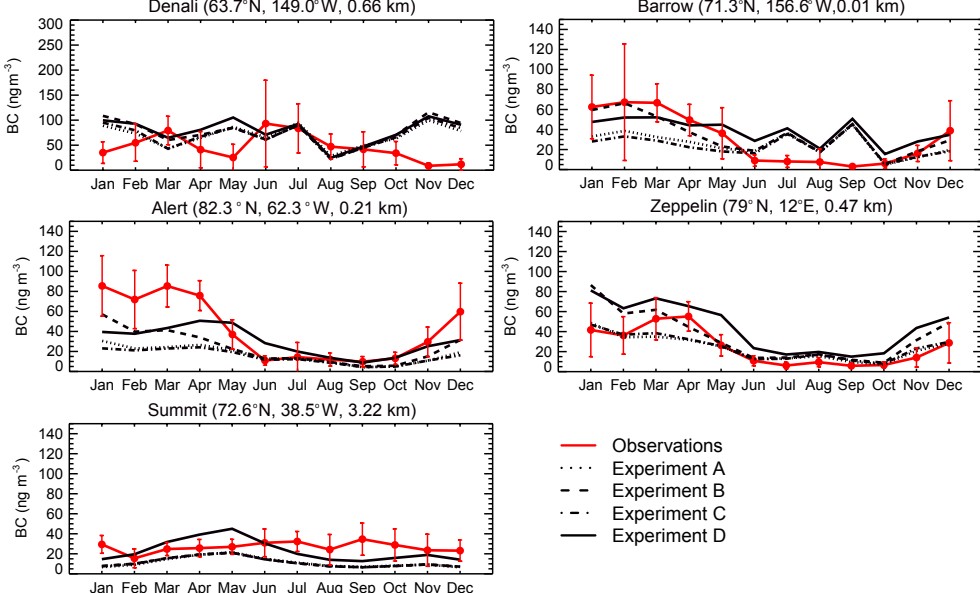

**Figure 5: Observed (red solid) and GEOS-Chem simulated (black curves: dotted−Exp. A, dashed−Exp. B, dash dotted−Exp. C, solid−Exp. D, see Table 2 and text for details) BC concentrations in air (ng m⁻³) at Denali, Barrow, Alert, Zeppelin, and Summit, averaged for 2007−09. Also shown are standard deviations of observations (error bars).**



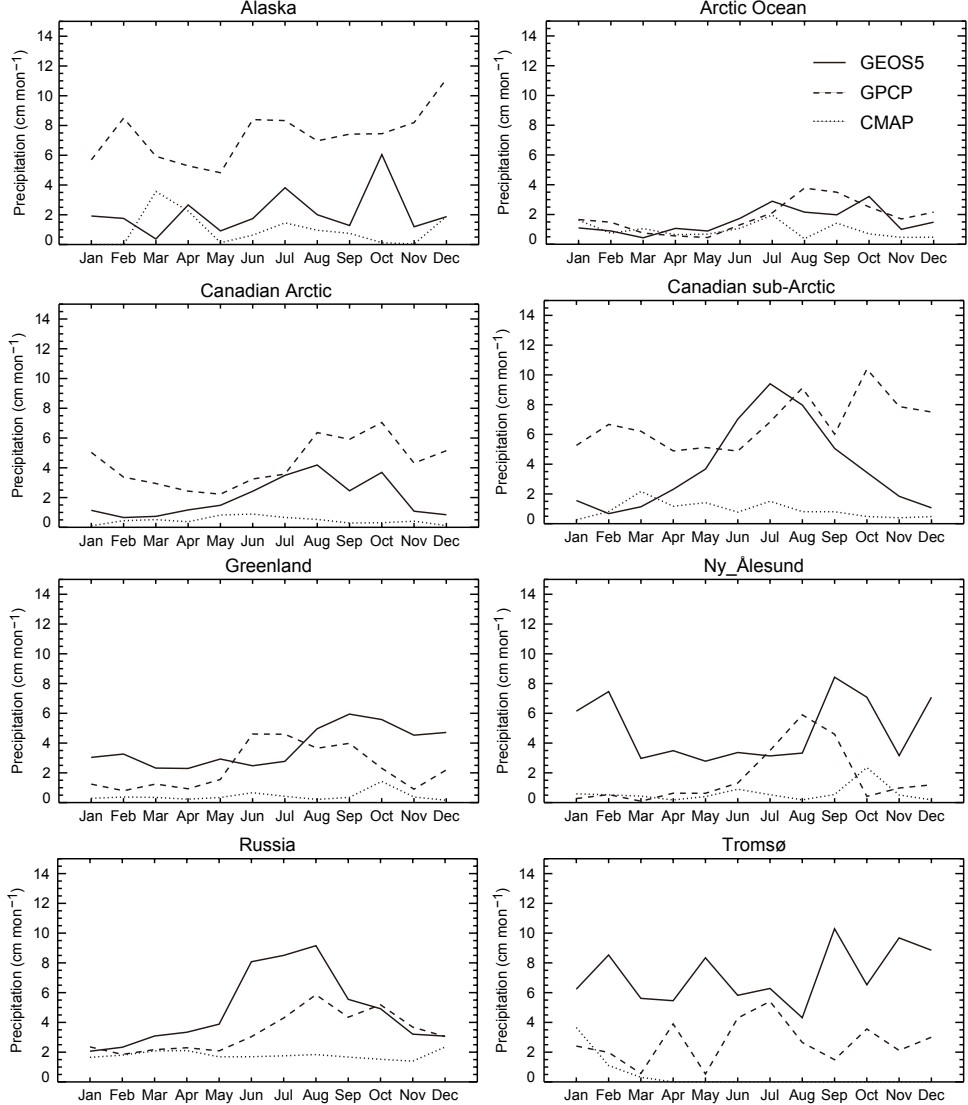

**Figure 6: Monthly precipitation (cm mon$^{-1}$) averaged over sub-regions in the Arctic for 2006−08 (Fig. 1). Data are from the Goddard Earth Observing System Model version 5 data assimilation system (GEOS-5 DAS), Global Precipitation Climatology Project (GPCP), and NOAA Climate Prediction Center Merged Analysis of Precipitation (CMAP).**




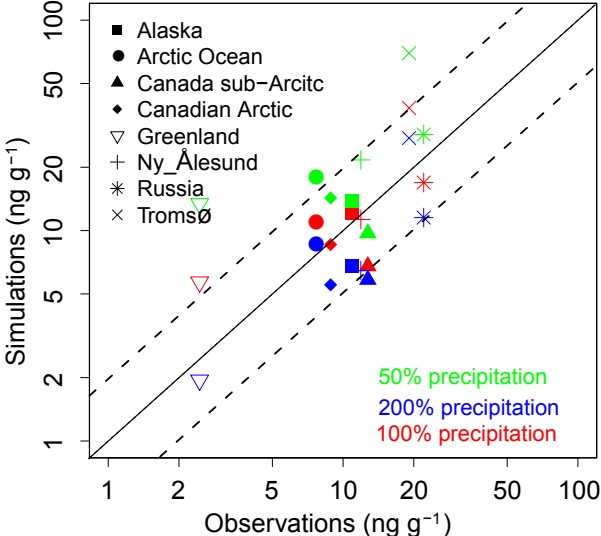

**Figure 7: Same as Fig. 3, but for Exp. D with standard precipitation (red symbols), 50% precipitation (green symbols), and 200% precipitation (blue symbols). See text for details.**



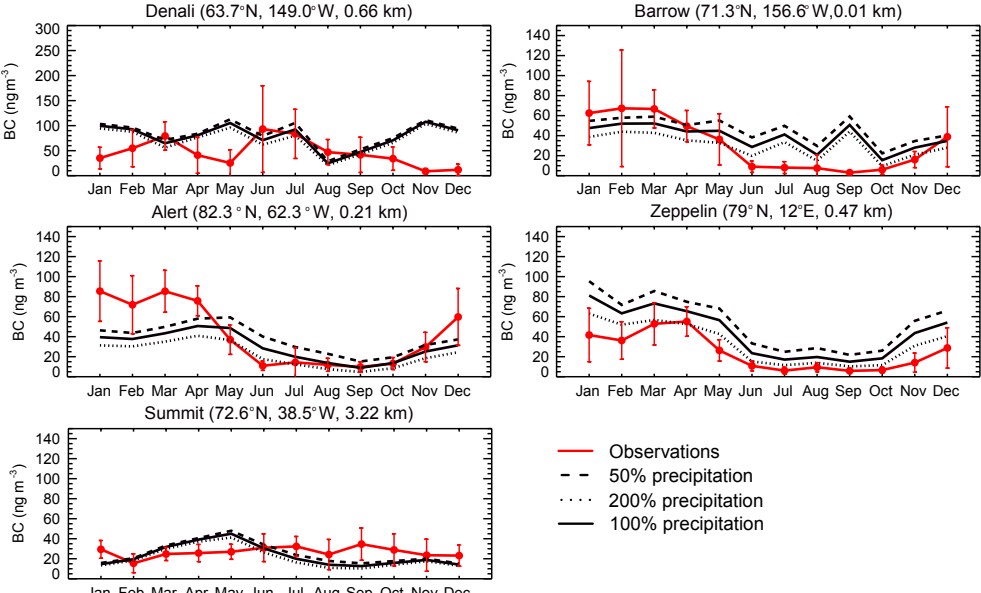

**Figure 8: Same as Fig. 5, but for Exp. D with standard precipitation (solid black), 50% precipitation (dashed black), and 200% precipitation (dotted black). See text for details.**