# Peer review of "Factors Controlling Black Carbon Distribution in the Arctic"

_Atmospheric Chemistry and Physics, 2016_

## Referee Comment (RC1) · A. Stohl (Referee) · 2 Oct 2016

This paper presents a model study of factors influencing BC distribution in the Arctic. The authors use a state-of-the-art model and perform several sensitivity studies to test how treatment of several model processes and emissions influence the simulation results. The study is largely solid, and the authors certainly know what they are doing. However, there are many ways in a complex model to influence the BC lifetime and simulated concentrations in remote areas. The authors only tested a few of them, and there needs to be some justification why exactly the selected ones are tested, while others are not touched upon. The authors also demonstrated some model improvements, but to what extent are they "right for the right reasons"? Couldn't some changes (including, for instance, increased emissions) just compensate errors somewhere else in the model? The comparison with the measurement data is not detailed enough to give process-specific information. Nevertheless, the paper could be published after some

considerable improvements, as also specified below.

The language in the paper often does not differentiate well enough between facts and model results. For instance, on page 10, line 1-2, it is said: "Our results show that WBF increases BC_snow by 20-80% in the eight Arctic sub-regions". In fact, the results show that, in the model, this is the case, but not that the WBF process in reality necessarily has the same effect. I think this reflects some over-confidence in the model which is not warranted. The wording should be such that the reader can distinguish between model-based findings, and findings that are more robust than just a model result. A similar example, not related to the model is "Gas flares are a rather small fraction of the global BC emissions but a dominant source in the Arctic". Again, this is not a fact but based on one study that used a highly uncertain emission factor. The language should reflect such uncertainties, e.g., "It has been suggested" or similar.

One problem of the paper is the use of the English language and, generally, a some-what sloppy style of writing. What I mean with the latter is, for instance, that references are not always written in the correct format (e.g., three examples on page 5, lines 28-29), or that there are quite many unnecessary mistakes like hyphenating within a line (page 5, line 23: John-son) or typos like "boundary layer eight" (Page 9, line 36). I suggest a much more careful checking of the manuscript, and a substantial improve-ment of the language. Numerous language errors like "The resulted BC…" (Page 9, line 3) make the paper difficult to read. I am sure there must be some native American speakers around at an American university who could help with that.

The Discussion section is more a repetition, rather than a real discussion. A discussion should go deeper, compare more with the existing literature, etc.

Page 2, ca. lines 32-35: It is true that models struggle reproducing Arctic BC. However, there have been recent improvements. For instance, the model intercomparison of Eckhardt et al. (2015) shows better results than the cited papers.

Page 11, middle paragraph: You discuss the differences it makes for snow radiative

forcing whether BC is deposited via dry deposition, or with wet deposition. However, is this really relevant? Processes within the snow are likely to redistribute the BC. Wouldn't this quickly eliminate any differences in the mode of BC deposition?

Minor:

Page 4, line 5: It is not really true that Barrow "receives minimal influence from anthropogenic effects". I assume you mean local emissions? There is quite some influence from the town of Barrow, which is partly filtered out based on wind sector analysis.

Page 4, line 36: Three stations are named, but only two MAC values are given.

Page 11, line 12: isentropically, not isotropically

Throughout the paper: It is not Ny_Ålesund, but Ny Ålesund

Page 13, line 23: latitude-longitude bands: What is that? Do you mean grid cells?

Page 13, line 2: boundary -> boundary layer?

Language (only a few examples):

Page 3, line 11: comprehensive measurements ***OF*** BC_snow

Page 4, line 4: through ***A*** filter

Page 4, line 5: It receives minimal. . .

Page 6, line 3: over snow and ice, ***AND*** found; delete AND)

Page 9, line 9: leads to significantLY reduction; remove "LY"

Page 9, lien 39: "when the stacks elevated BC emissions to the free troposphere": What does that mean?

Page 15, line 19: could -> cloud

Page 15, line 31: AeroComp -> Aerocom

Page 16, line 10: in Western Russia IS biased high.

Page 16, line 30: decrease causeD by

Reference:

Eckhardt, S., et al. (2015): Current model capabilities for simulating black carbon and sulfate concentrations in the Arctic atmosphere: a multi-model evaluation using a comprehensive measurement data set. Atmos. Chem. Phys. 15, 9413-9433, doi:10.5194/acp-15-9413-2015.

---

## Referee Comment (RC2) · Anonymous Referee #2 · 17 Oct 2016

In this study, the authors investigate the importance of three processes (Arctic emissions, dry deposition, and aerosol mass transfer from cloud water in ice clouds) to improve the comparison between black carbon concentrations simulated by the chemistry-transport model GEOS-Chem and Arctic observations. They find that emitting black carbon directly in the Arctic is more efficient than slowing down deposition during transport, although reducing the efficiency of ice-cloud scavenging also helps to improve the model.

The paper is generally well-written, although there are instances of plagiarism that must be removed, and the discussion and conclusion sections are not adequate and need to be rewritten. Figures are good, and Table 3 is a particularly interesting literature review that will be useful to other modellers. Possible implications for radiative forcing mentioned in section 4.2 are interesting, although they remain speculative. Scientifically, the study's weaker points are an unclear motivation for the choice of processes to be

studied, and a very underused discussion on precipitation biases.

I recommend publication after the comments below are addressed. Those comments are aimed at improving the discussion, but I do not expect that further analysis will be required.

**1 Main comments**

- There are too many instances of plagiarism in the paper, where the authors quote directly from sources without making that fact clear. Page 4, lines 7–8 have been taken directly from the description of the Barrow site at www.esrl.noaa.gov/gmd/obop/brw. Page 5, lines 19–20 are taken from Elvidge et al. (2009). The sentence that follows seems to have been taken from a World Bank document. The sentence after that (lines 21-23) is taken from Stohl et al. (2013), complete with hyphenation! It is important that authors realise that citing a paper does not mean taking whole statements from it: one must rephrase and extract the information that is relevant to the present study. I have not looked further for instances of potential plagiarism, but the authors need to remove them when revising their paper.

- The study often reads like a model tuning exercise: the authors have acted on three processes but the reader is left piecing together why those three processes have been targeted in priority. The information is already there to some extent in sections 3.2, 3.3, and 3.4, but would benefit from being gathered in a motivation paragraph in the introduction. Such a paragraph could also discuss why the authors believe that aspects like emission injection heights (page 9 from line 35), size distribution, or absorption properties have not been varied to improve simulated Arctic concentrations.

  Then, the authors seem to stop improving their model once its simulation is within

a factor 2 of the observations (page 9, line 3). It feels arbitrary, but I acknowledge that such practice is common in global modelling. Still, a short discussion would be useful in section 5. Schutgens et al. (2016) could be a relevant reference there.

• Section 4.4 is really interesting. Its key message is that difficulties in modelling BC concentrations in the Arctic may be really dominated by non-aerosol aspects of the model. It is a sobering conclusion: why put so much effort in having better aerosol emissions and physical processes when errors in precipitation can wipe out improvements on the aerosol side? Considering its interest, it is surprising that the findings of section 4.4 are not mentioned in the abstract. The section needs to be much better integrated in the paper.

• The discussion section is not adequate. First, the two paragraphs on page 15 lines 9–20 and 28–35 clearly do not belong there because they hardly mention the present study. Instead, we expect to see in that section the answers to questions like: What have we learned that is new? Are results consistent with previous studies? If not, why should we have more confidence in the authors' results than previous results?

Similarly, section 6 is not an adequate conclusion. It gives too many quantitative details when it really should focus on the key messages and list the limitations of the study.

**2  Other comments**

• Page 1, lines 34–35: That sentence reads like the nuclear war described by Warren and Wiscombe (1985) actually took place. Fortunately, that was only a model study and the sentence should be rewritten to clarify that point.

- Page 2, lines 5 and 6: Are the authors comparing Arctic-averaged BC radiative forcing to globally-averaged ozone radiative forcing? If so, that comparison would be misleading and should be removed.

- Page 6, lines 10–11: If I understand well, the model already calculates dry deposition velocities using an analogy with resistances in series. The authors only updated the values used to represent the surface. Is that correct? If so, that statement should be clarified.

- Page 8, lines 3–6: That procedure is unclear. Can it be clarified?

- Page 13, line 27: How is precipitation halved or doubled in the model? By scaling precipitation rates? Does that method cause an imbalance in the simulated water budget?

- Page 26, Table 1: How is "data availability" calculated?

**3 Technical comments**

- Throughout the paper: Ny_Ålesund should be written Ny-Ålesund.

- Page 4, line 5: typo "It is received"

- Page 9, line 9: rephrase to "significant reduction".

- Page 11, line 8: rephrase to "thereby reducing"

- Page 14, line 3: less BC particles → fewer BC particles.

**4 References**

Schutgens, N. A. J., Gryspeerdt, E., Weigum, N., Tsyro, S., Goto, D., Schulz, M., and Stier, P.: Will a perfect model agree with perfect observations? The impact of spatial sampling, *Atmos. Chem. Phys.*, 16, 6335-6353, doi:10.5194/acp-16-6335-2016, 2016.

---

## Author Comment (AC1) · 16 Dec 2016

**Referee #1**

**Major Comments:**

*"This paper presents a model study of factors influencing BC distribution in the Arctic. The authors use a state-of-the-art model and perform several sensitivity studies to test how treatment of several model processes and emissions influence the simulation results. The study is largely solid, and the authors certainly know what they are doing."*

1 *"There are many ways in a complex model to influence the BC lifetime and simulated concentrations in remote areas. The authors only tested a few of them, and there needs to be some justification why exactly the selected ones are tested, while others are not touched upon."*

**Response**: Points well taken. We've re-written the introduction (Page 3, Lines 1–29 in the track-change manuscript) to justify focusing on the selected processes in this study.

2 *"The authors also demonstrated some model improvements, but to what extent are they "right for the right reasons"? Couldn't some changes (including, for instance, increased emissions) just compensate errors somewhere else in the model? The comparison with the measurement data is not detailed enough to give process-specific information."*

**Response**: Points well taken. We now acknowledge as much in the text. Clearly the above comments are applicable to most if not all modelling exercises given the nature of complex, intertwining processes in models. Constraining individual processes of BC is often challenging. As such, our focus is more geared toward highlighting missing processes or ones that were previously unaccounted for in governing BC in the Arctic. We try our best to constrain the processes we discussed in the manuscript using process-specific observations and related information.

1) Gas flaring emissions in the Arctic.
To our knowledge, there are no measurements of BC emission factors of gas flares in Russia so far. We discussed the large uncertainties of BC emission factors from gas flares in Sect. 4.1 on Page 9, Lines 33–37 and Page 10, Lines 1–7. We also use $BC_{snow}$ measurements near the gas flares and $BC_{air}$ measurements at Zeppelin, the closest in-situ measurement station to the gas flares, to constrain flaring emissions in the same Section (Page 10, Lines 27–32).

2) Dry deposition velocity over snow and ice.
We collected and compared available observations of aerosol dry deposition velocities in the Arctic with model simulations. The model results are in broad agreement with observations. See Table 3 and the first paragraph of Sect. 4.2 for details.

3) Wegener-Bergeron-Findeisen (WBF) process in mixed-phase clouds.

Effects of WBF on global BC distribution are presented in a companion paper Qi et al., (2016) under review in ACPD (#acp-2016-706). Detailed validation of this process in the Arctic is also discussed in that paper. We clarified this in the text (Page 12, Line 15).

3 *"The language in the paper often does not differentiate well enough between facts and model results. For instance, on page 10, line 1-2, it is said: "Our results show that WBF increases BC_snow by 20-80% in the eight Arctic sub-regions". In fact, the results show that, in the model, this is the case, but not that the WBF process in reality necessarily has the same effect. I think this reflects some over-confidence in the model which is not warranted. The wording should be such that the reader can distinguish between model-based findings, and findings that are more robust than just a model result. A similar example, not related to the model is "Gas flares are a rather small fraction of the global BC emissions but a dominant source in the Arctic". Again, this is not a fact but based on one study that used a highly uncertain emission factor. The language should reflect such uncertainties, e.g., "It has been suggested" or similar."*

**Response**: Revised accordingly. We've also evaluated other statements in the manuscript and differentiated facts and model results.

4 *"One problem of the paper is the use of the English language and, generally, a some-what sloppy style of writing. What I mean with the latter is, for instance, that references are not always written in the correct format (e.g., three examples on page 5, lines 28-29), or that there are quite many unnecessary mistakes like hyphenating within a line (page 5, line 23: John-son) or typos like "boundary layer eight" (Page 9, line 36). I suggest a much more careful checking of the manuscript, and a substantial improvement of the language. Numerous language errors like "The resulted BC. . ." (Page 9, line 3) make the paper difficult to read. I am sure there must be some native American speakers around at an American university who could help with that."*

**Response**: Revised as suggested here and in the 'Minor comments' below. A native English speaker helped us to proofread the manuscript as well.

5 *"The Discussion section is more a repetition, rather than a real discussion. A discussion should go deeper, compare more with the existing literature, etc."*

**Response**: We've completely re-written the discussions section.

6. *Page 2, ca. lines 32-35: It is true that models struggle reproducing Arctic BC. However, there have been recent improvements. For instance, the model intercomparison of Eckhardt et al. (2015) shows better results than the cited papers.*

**Response**: Agree. We have added discussions of Eckhardt et al. (2015) on Page 2, Lines 36–39 as follows:

"Specifically, mean $BC_{air}$ during January to March was underestimated by about a factor of 2 for the mean of all models, although the discrepancy is up to a factor of 27 for individual models (Eckhardt et al., 2015)"

*7. Page 11, middle paragraph: You discuss the differences it makes for snow radiative forcing whether BC is deposited via dry deposition, or with wet deposition. However, is this really relevant? Processes within the snow are likely to redistribute the BC. Wouldn't this quickly eliminate any differences in the mode of BC deposition?*

**Response**: Excellent point. We've added discussion accordingly on Page 11, Lines 35–40 as follows:

"This effect is critical before melting season, because melting might quickly eliminates the differences in the mode of BC deposition. Other post-depositional processes include wind-driven drifting and sublimation (Doherty et al., 2013). The former does not change the fraction of external and internal mixing of BC with snow. The later might expose BC particles in the internally mixed BC/snow composite out and reduce the fraction of internally mixed BC/snow composite. Yet this process occurs slowly in a relatively long time."

**Minor:**

1. *"Page 4, line 5: It is not really true that Barrow "receives minimal influence from anthropogenic effects". I assume you mean local emissions? There is quite some influence from the town of Barrow, which is partly filtered out based on wind sector analysis."*

**Response**: We meant "local anthropogenic emissions". Fixed.

2. *"Page 4, line 36: Three stations are named, but only two MAC values are given."*

**Response**: Fixed. We have added the third MAC value.

3. *"Page 11, line 12: isentropically, not isotropically"*

**Response**: Fixed.

4. *"Throughout the paper: It is not Ny_Ålesund, but Ny Ålesund*

**Response**: We changed "Ny_Ålesund" to "Ny-Ålesund" as suggested by the other reviewer and as used in many other literatures (Eleftheriadis et al., 2009; Doherty et al., 2010; Sharma et al., 2013; Yttri et al., 2014)

5. *"Page 13, line 23: latitude-longitude bands: What is that? Do you mean grid cells?"*

**Response**: It is grid cells. Fixed.

6. *"Page 13, line 2: boundary -> boundary layer?"*

**Response**: Fixed.

*Language (only a few examples):*

7. *"Page 3, line 11: comprehensive measurements \*\*\*OF\*\*\* BC_snow"*

**Response**: Fixed.

8. *"Page 4, line 4: through \*\*\*A\*\*\* filter"*

**Response**: Fixed.

9. *"Page 4, line 5: It receives minimal. . ."*

**Response**: Fixed.

10. *"Page 6, line 3: over snow and ice, \*\*\*AND\*\*\* found; delete AND)"*

**Response**: Fixed.

11. *"Page 9, line 9: leads to significantLY reduction; remove "LY""*

**Response**: Fixed.

12. *"Page 9, lien 39: "when the stacks elevated BC emissions to the free troposphere": What does that mean?"*

**Response**: We modified it to "when the stacks emitted BC to the free troposphere"

13. *"Page 15, line 19: could -> cloud"*

**Response**: Fixed.

14. *"Page 15, line 31: AeroComp -> Aerocom"*

**Response**: Fixed.

15. *"Page 16, line 10: in Western Russia IS biased high."*

**Response**: Fixed.

16. *"Page 16, line 30: decrease causeD by"*

**Response**: Fixed.

[revised manuscript text omitted]

---

## Author Comment (AC2) · 16 Dec 2016

**Referee #2**

*Main comments*

1. *"There are too many instances of plagiarism in the paper, where the authors quote directly from sources without making that fact clear. Page 4, lines 7–8 have been taken directly from the description of the Barrow site at www.esrl.noaa.gov/gmd/obop/brw. Page 5, lines 19–20 are taken from Elvidge et al. (2009). The sentence that follows seems to have been taken from a World Bank document. The sentence after that (lines 21-23) is taken from Stohl et al. (2013), complete with hyphenation! It is important that authors realize that citing a paper does not mean taking whole statements from it: one must rephrase and extract the information that is relevant to the present study. I have not looked further for instances of potential plagiarism, but the authors need to remove them when revising their paper."*

**Response**: Thanks for pointing out the problem. We've revised these sentences and cited previous studies properly.

2. *"The study often reads like a model tuning exercise: the authors have acted on three processes but the reader is left piecing together why those three processes have been targeted in priority. The information is already there to some extent in sections 3.2, 3.3, and 3.4, but would benefit from being gathered in a motivation paragraph in the introduction. Such a paragraph could also discuss why the authors believe that aspects like emission injection heights (page 9 from line 35), size distribution, or absorption properties have not been varied to improve simulated Arctic concentrations."*

**Response**: Points well taken. We have added proper justifications and discussions in the introduction to clarify and highlight why we investigated these processes (Page 3, Lines 1–29).

3. *"Then, the authors seem to stop improving their model once its simulation is within a factor 2 of the observations (page 9, line 3). It feels arbitrary, but I acknowledge that such practice is common in global modelling. Still, a short discussion would be useful in section 5." Schutgens et al. (2016) could be a relevant reference there."*

**Response**: Thanks for introducing Schutgens et al. (2016) to us. We investigated several key factors affecting BC distribution in the Arctic and tried our best to improve model simulations by using observational and modeling information available to us. In this

study, as a big step forward, we significantly reduced model biases to a factor of two of the observations. Since the effects of different factors on BC simulations in the Arctic are rather complex, we are unable to make the model perfect through only one study. Further model improvements will be investigated in our future study. In addition, Schutgens et al. (2016) quantified the error due to different spatial sampling of global models and point observations, which is up to 160%. Thus, the model-observation discrepancy is acceptable for global models in this study. We've added the corresponding discussions on Page 9, Lines 12–14 as follows:

"This discrepancy is acceptable for global models because it has been suggested that the error due to different spatial sampling of global models (~200 km) and point observations is up to 160% (Schutgens et al., 2016)."

4. *"Section 4.4 is really interesting. Its key message is that difficulties in modelling BC concentrations in the Arctic may be really dominated by non-aerosol aspects of the model. It is a sobering conclusion: why put so much effort in having better aerosol emissions and physical processes when errors in precipitation can wipe out improvements on the aerosol side? Considering its interest, it is surprising that the findings of section 4.4 are not mentioned in the abstract. The section needs to be much better integrated in the paper."*

**Response**: Points well taken. We've summarized this result in the abstract on Page 1, Lines 27–29 as follows:

"In addition, we find that the poorly constrained precipitation in the Arctic may introduce large uncertainties in estimating BC$_{snow}$. Doubling (halving) precipitation introduces a positive (negative) bias similar to the magnitude of the overall effects of flaring emissions and the WBF effect."

5. *"The discussion section is not adequate. First, the two paragraphs on page 15 lines 9–20 and 28–35 clearly do not belong there because they hardly mention the present study. Instead, we expect to see in that section the answers to questions like: What have we learned that is new? Are results consistent with previous studies? If not, why should we have more confidence in the authors' results than previous results?*

**Response**: Points well taken. We've re-written the discussion part completely.

6. *Section 6 is not an adequate conclusion. It gives too many quantitative details when it really should focus on the key messages and list the limitations of the study."*

**Response**: Points well taken. We've completely re-written the conclusion part (Sect. 6) accordingly.

*Other comments*

1. *"Page 1, lines 34–35: That sentence reads like the nuclear war described by Warren and Wiscombe (1985) actually took place. Fortunately, that was only a model study and the sentence should be rewritten to clarify that point."*

**Response**: Agree. We revised the sentence accordingly on Page 1, Line 36 and Page 2, Line 1 as follows:

"Warren and Wiscombe (1985) highlighted the climate effect of fallen soot from 'smokes' for an assumed nuclear war scenario, which reduced the surface reflectivity of snow and sea ice in the Arctic."

2. *"Page 2, lines 5 and 6: Are the authors comparing Arctic-averaged BC radiative forcing to globally-averaged ozone radiative forcing? If so, that comparison would be misleading and should be removed."*

**Response**: We used ozone radiative forcing averaged for the Arctic. We rephrased the sentence (Page 2, Line 8) to clarify as "… comparable to the forcing of tropospheric ozone in springtime Arctic (0.34 W m$^{-2}$, Quinn et al., 2008)."

3. *"Page 6, lines 10–11: If I understand well, the model already calculates dry deposition velocities using an analogy with resistances in series. The authors only updated the values used to represent the surface. Is that correct? If so, that statement should be clarified."*

**Response**: The model uses a uniform and constant $v_d$ value of 0.03 cm s$^{-1}$ for dry deposition velocity of aerosols over snow and ice. We use the resistence-in-series method to estimate $v_d$ of BC over snow and ice in this study. We revised the sentence as

"We apply the resistance-in-series method to calculate $v_d$ of BC over snow and ice, replacing the uniform $v_d$ of 0.03 cm s$^{-1}$."

4. *"Page 8, lines 3–6: That procedure is unclear. Can it be clarified?"*

**Response**: We have clarified the procedure on Page 8, Lines 14–20 as follows:

"The top and bottom snow depths of each sample and the collection date are provided in the observation dataset (Doherty et al., 2010). We accumulate snow precipitation (GEOS-5) in the model from the collection date backward until the modelled snow depths, respectively, reach the observed top and bottom depths of the snow sample, then the two dates are stored. We use the average BC deposition fluxes and snow precipitation between the two dates to estimate the $BC_{snow}$ for the sample. The rate of snow accumulation at the surface is estimated as snow precipitation flux ($kg\ m^{-2}\ s^{-1}$) over snow density ($kg\ m^{-3}$). The observed annual average snow density is 300 $kg\ m^{-3}$ over the Arctic basin, increasing from 250 $kg\ m^{-3}$ in September to 320 $kg\ m^{-3}$ in May with little geographical variation across the Arctic (Warren et al., 1999; Forsström et al., 2013). We use the annual average snow density in the estimate."

5. *"Page 13, line 27: How is precipitation halved or doubled in the model? By scaling precipitation rates? Does that method cause an imbalance in the simulated water budget?"*

**Response**: We doubled and halved precipitation by scaling the precipitation rate. We use a chemical transport model in this study. It is driven by a reanalysis meteorological data from GEOS-5. Thus, changing precipitation rate does not affect other water-related variables or balance of water budget in the model.

6. *"Page 26, Table 1: How is "data availability" calculated?"*

**Response**: It is estimated as the ratio of available data to the sum of available and missing data. We clarified this in the note of Table 1.

*Technical comments*

1. *"Throughout the paper: Ny_Ålesund should be written Ny-Ålesund."*

**Response**: Fixed.

2. *"Page 4, line 5: typo "It is received""*

**Response**: Fixed.

3. *"Page 9, line 9: rephrase to "significant reduction""*

**Response**: Fixed.

4. *"Page 11, line 8: rephrase to "thereby reducing""*

**Response**: Fixed.

5. *"Page 14, line 3: less BC particles → fewer BC particles."*

**Response**: Fixed.

[revised manuscript text omitted]